# Amyloid fibril structures link CHCHD10 and CHCHD2 to neurodegeneration

Guohua Lv[1,6], Nicole M. Sayles[2,6], Yun Huang [3,4,6], Chiara Mancinelli [1,6], Kevin McAvoy [2], Neil A. Shneider[5], Giovanni Manfredi[2], Hibiki Kawamata [2] ✉ & David Eliezer [1,2] ✉

Mitochondrial proteins CHCHD10 and CHCHD2 are mutated in rare cases of heritable FTD, ALS and PD and aggregate in tissues affected by these diseases. Here, we show that both proteins form amyloid fibrils and report cryo-EM structures of fibrils formed from their disordered N-terminal domains. The ordered cores of these fibrils are comprised of a region highly conserved between the two proteins, and a subset of the CHCHD10 and CHCHD2 fibril structures share structural similarities and appear compatible with sequence variations in this region. In contrast, disease-associated mutations p.S59L in CHCHD10 and p.T61I in CHCHD2, situated within the ordered cores of these fibrils, cannot be accommodated by the wildtype structures and promote different protofilament folds and fibril structures. These results link CHCHD10 and CHCHD2 amyloid fibrils to neurodegeneration and further suggest that fibril formation by the WT proteins could also be involved in disease etiology.

Coiled-coil-helix-coiled-coil-helix domain containing 10 (CHCHD10, D10) is the first mitochondrial protein to be associated with familial frontotemporal dementia (FTD) and amyotrophic lateral sclerosis (ALS)[1,2]. Numerous pathogenic D10 variants have been reported[3], including the first mutation to be identified, p.S59L[1]. D10 forms inclusions in diseased neurons[4] and mutant p.S55L (equivalent of human p.S59L) D10 knock-in (KI) mice (D10^S55L) develop toxic protein aggregates in multiple tissues, resulting in neurological defects, myopathy, and cardiomyopathy[5,6], but the nature and structure of these aggregates remain unknown. D10 localizes to the intermembrane space of mitochondria and associates with the mitochondrial inner membrane[7], as well as other mitochondrial proteins[7–9], including its paralog protein CHCHD2 (D2). As for D10, mutations in D2 are associated with neurodegenerative diseases, including Parkinson's disease (PD), dementia with Lewy bodies (DLB), multiple system atrophy (MSA), and Alzheimer's disease (AD)[10–14].

Although the normal functions of both D10 and D2 and their potential relation to neurodegeneration and disease remain poorly understood, early studies suggested that loss of function may play a role in disease etiology[15,16]. However, more recent work has shown that constitutive genetic ablation of D10[5,17] or D2[18] in mice does not appear to impair mitochondrial functions or cause pathological consequences. Knockout (KO) of both D10 and D2 causes mitochondrial alterations[17], but only very mild disease phenotypes, suggesting that the proteins can functionally complement each other in single KO and are not essential for life, even in double KO. Combined with the autosomal dominant nature of disease-associated D10 mutations, these observations provide strong support for a toxic gain-of-function mechanism of the mutant proteins. The proteotoxic mechanisms remain to be fully elucidated, but in animal models, disease phenotypes correlate with protein aggregation and deposition, which causes structural damage in mitochondria, accompanied by mitochondrial integrated stress response activation and profound metabolic rewiring in affected tissues[17,19]. In this study, to better understand the potential role of D10 and D2 aggregates in disease, we investigated their structural properties.

[1]Department of Biochemistry, Weill Cornell Medicine, 1300 York Avenue, New York, NY, USA. [2]Feil Family Brain and Mind Research Institute, Weill Cornell Medicine, New York, NY, USA. [3]Department of Physiology & Biophysics, Weill Cornell Medicine, New York, NY, USA. [4]Howard Hughes Medical Institute, Chevy Chase, MD, USA. [5]Department of Neurology, Center for Motor Neuron Biology and Disease, Columbia University Irving Medical Center, New York, NY, USA. [6]These authors contributed equally: Guohua Lv, Nicole M. Sayles, Yun Huang, Chiara Mancinelli. ✉e-mail: hik2004@med.cornell.edu; dae2005@med.cornell.edu

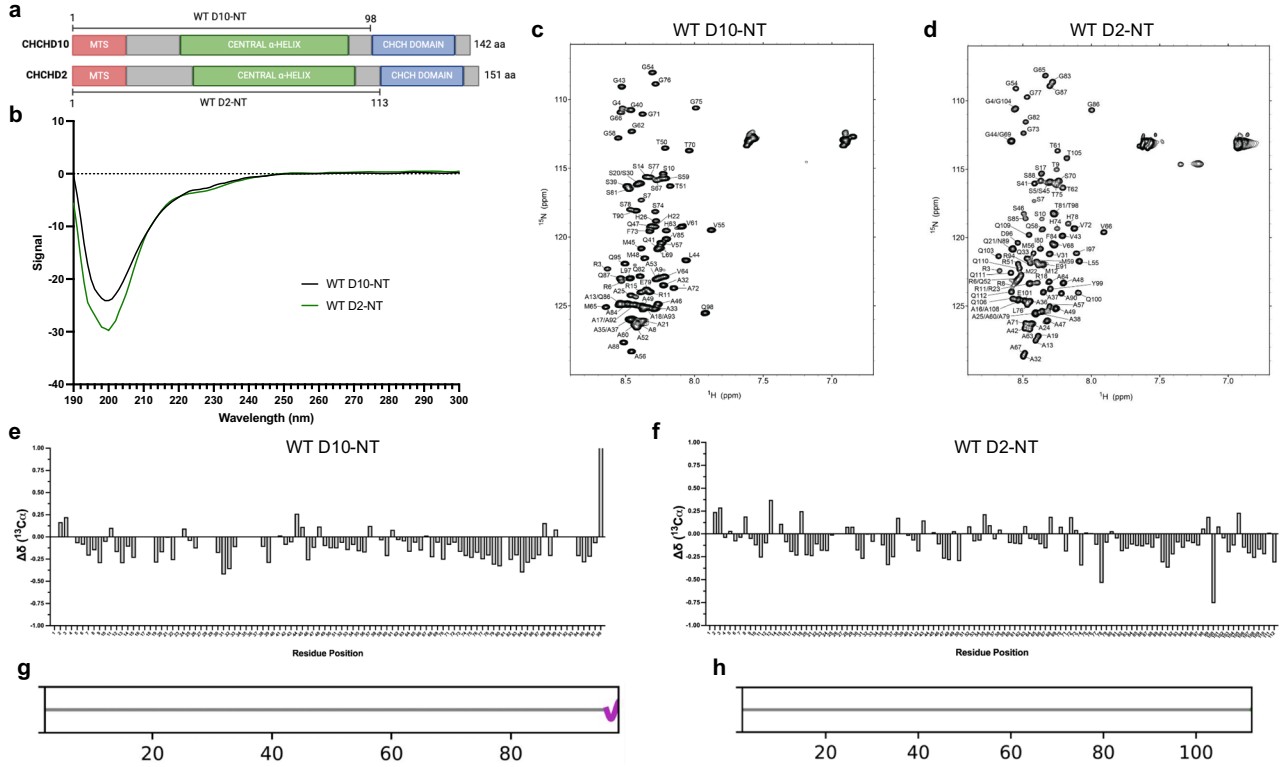

Fig. 1 | The N-terminal domains of D10 and D2 are disordered in solution.
a Schematic of full-length human CHCHD10 and CHCHD2 delineating the putative mitochondrial targeting sequence (MTS, red), central α-helix domain (green), and the coiled-coil-helix-coiled-coil-helix (CHCH) domain (blue). The N-terminal (NT) fragments used in this study are indicated. b CD spectra of WT D10-NT (black) and WT D2-NT (green). c, d NMR $^{15}$N-$^{1}$H HSQC spectra of WT D10-NT (c) and WT D2-NT (d) annotated with backbone resonance

assignments (see Supplementary Figs. 15 and 16 for full-page versions). e, f Secondary carbon chemical shifts of WT D10-NT (e) and WT D2-NT (f).
g, h Graphical representation of secondary structure determined by CheSPI for WT D10-NT (g) and WT D2-NT (h). The horizontal grey bar indicates a random coil (no secondary structure). The magenta sinusoidal line for D10-NT indicates the presence of one turn of a 3–10 helix for the final three C-terminal residues. Source data for b, e, and f are provided as a Source Data file.

The N-terminal regions of D10 (residues 1–100) and D2 (residues 1–112), which precede their folded eponymous CHCH domains, resemble low complexity prion-like domains that form the core of amyloid aggregates of other FTD/ALS-associated proteins such as TDP-43, FUS, and hnRNPA1[20–24]. Over 50% of both N-terminal sequences are comprised of just 4 amino acids, alanine, proline, serine, and glycine. Most disease-associated D10 and D2 mutations, including D10 p.S59L and p.R15L and D2 p.T61I and p.V66M, fall within these N-terminal regions[3,25]. Despite the presence of these low complexity domains, the potential for amyloid fibril formation by D10 and/or D2 has not been investigated.

Here, we demonstrate that the low complexity N-terminal domains of both D10 and D2, as well as the full-length proteins, can form amyloid fibrils in vitro and in vivo. We solved the structures of N-terminal domain fibrils using cryogenic electron microscopy (cryo-EM) and show that they are comprised of the most highly conserved region between the two proteins. We further show that disease-associated mutations within this region promote alternative fibril structures. Together, our findings strongly implicate amyloid fibril formation in the etiology of D10- and D2-associated neurodegenerative disease.

## Results

### The N-terminal low complexity domains of D10 and D2 are disordered

The CHCH domains of D10 and D2 are predicted to be folded, but the structural properties of the low complexity domains of both proteins, which include putative mitochondrial targeting sequences (MTS) and central α-helix domains (Fig. 1a), have been explored

primarily through computational simulations[26] and modeling[7]. To assess experimentally whether these domains contain any structure, we expressed and purified recombinant N-terminal fragments comprising D10 residues 1–98 (D10-NT) and D2 residues 1–113 (D2-NT) (Fig. 1a) and obtained their circular dichroism (CD) and nuclear magnetic resonance (NMR) spectra. CD spectra for both fragments were characteristic of disordered polypeptides, with a single minimum around 200 nm (Fig. 1b). Similarly, NMR proton–nitrogen correlation spectra of each fragment exhibited limited dispersion in the proton dimension and clustering of signals in the $^{15}$N dimension according to residue type, also characteristic of disordered proteins (Fig. 1c, d). To obtain additional information regarding residual secondary structure, we assigned the backbone NMR resonances of both constructs and assessed secondary structure using chemical shifts (Fig. 1e, f). For both fragments, chemical shift deviations from random coil values were minimal, indicating a lack of secondary structure. Analysis of all assigned chemical shifts using CheSPI[27] confirmed this result (Fig. 1g, h).

### The N-terminal low complexity domains of D10 and D2 form amyloid fibrils

To assess whether D10-NT and D2-NT can form amyloid fibrils, as is the case for many other low complexity prion-like domains associated with neurodegenerative diseases, we agitated both fragments in solution and assayed for amyloid fibril formation using Thioflavin T (ThT) fluorescence and transmission electron microscopy (TEM). We observed a time-dependent increase of ThT fluorescence for both constructs, with D10 exhibiting faster kinetics (Fig. 2a, b). Both reactions, when imaged by TEM, gave rise to the formation of filamentous

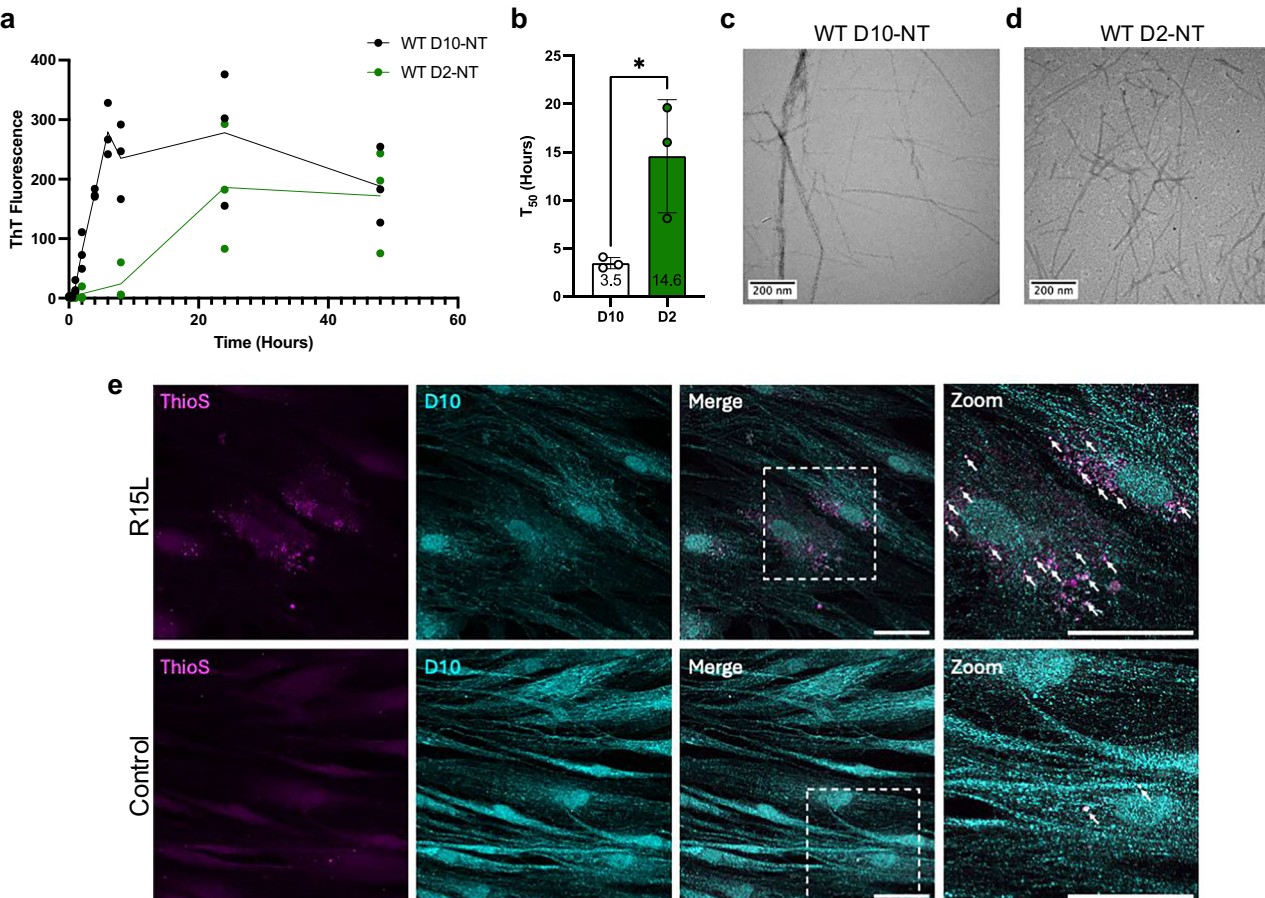

**Fig. 2 | The N-terminal domains of D10 and D2 form amyloid fibrils. a** ThT monitored aggregation of purified recombinant D10-NT (black) and D2-NT (green). **b** Time to reach half maximal ThT fluorescence ($T_{50}$) for D10-NT and D2-NT assembly (mean ± SD, $n = 3$ biological replicates, unpaired two-tailed para-mettric Student's $t$-test; *$p = 0.031$). **c** TEM micrograph of D10-NT agitated at ca. 20 μM concentration for 24 h at 1000 RPM. Similar results were obtained for multiple (>2) independent samples/experiments. **d** TEM micrograph of D2-NT agitated at ca. 80 μM concentration for 192 h at 1200 RPM. Results were obtained from a single sample/experiment but are corroborated by cryo-EM images of independent samples. **e** Skin fibroblasts from p.R15L D10 patients and controls immunostained for D10 (cyan) and co-stained with ThioS (magenta), showing colocalization (arrows). Scale bars: 50 μm. Similar results were obtained from a second independent sample/experiment. Source data for **a** and **b** are provided as a Source Data file.

species resembling classic amyloid fibrils (Fig. 2c, d). Next, we sought to determine if D10 amyloid could be detected in patients carrying disease-causing mutations. Although such patients are quite rare, we were able to obtain skin fibroblasts from patients with the p.R15L disease-associated D10 mutation. Staining of these cells with the amyloid-specific dye Thioflavin S (ThioS) revealed ThioS-positive punctate regions that were co-labeled by an antibody against D10 (Fig. 2e), indicating that these cells contain amyloid aggregates, likely containing D10.

To further assess whether D10 forms amyloid fibrils in vivo, we extracted sarcosyl insoluble material from D10^SSSL mice, which reca-pitulate D10 deposition as well as disease-associated phenotypes[5]. Mouse D10 is highly homologous to human D10 (Fig. 3a), with serine residue 55 corresponding to human residue S59, which is mutated to leucine in disease. The presence of D10 aggregates in these extracts, but not in extracts from WT mice (D10^WT), was confirmed using a filter trap assay (Fig. 3b). TEM imaging following D10-immunogold staining of extracts revealed gold-labeled fibrillar species consistent with amyloid fibrils (Fig. 3c), which were absent from D10^WT extracts. Gold labeling was sparse and tended to occur at the ends of the observed fibrils, suggesting that the antibody epitope, which is not reported, may be at least partly occluded by the fibril structure. To assess this, we also performed immunogold staining of in vitro assembled D10-NT fibrils using the same antibody. Indeed, the sparsity and pattern of

staining of the in vitro fibrils (Fig. 3d) resembled that observed for the D10^SSSL extracts. Given that cleavage of D10 and D2 has not been reported, in vivo fibrils could consist of full-length D10 protein. We therefore investigated whether recombinant full-length D10 and D2 could also form amyloid fibrils in vitro. Incubation of both full-length proteins with agitation resulted in increased ThT fluorescence (Sup-plementary Fig. 1a, b), as observed for D10-NT and D2-NT. Cryo-EM micrographs of both reactions revealed species resembling classic amyloid fibrils (Supplementary Fig. 1c, d). The fibrils appeared to be decorated with punctate densities, possibly corresponding to the fol-ded CHCH domains of both proteins, suggesting that fibril formation of the full-length proteins is still mediated by the N-terminal domains.

### The structured core of D10-NT and D2-NT amyloid fibrils is conserved between D10 and D2

Using cryo-EM, we observed several distinct D10-NT fibril polymorphs (see Methods section on Cryo-EM sample preparation and data col-lection) and determined three different structures. The formation of multiple different polymorphs is commonly observed for in vitro assembled amyloid fibrils[28]. Different polymorphs can result from even slight changes in sample conditions, such as pH and buffer composition[28,29], and can in principle represent both different end-points of an aggregation reaction as well as different time points along the aggregation process[30]. The first structure we obtained (polymorph-

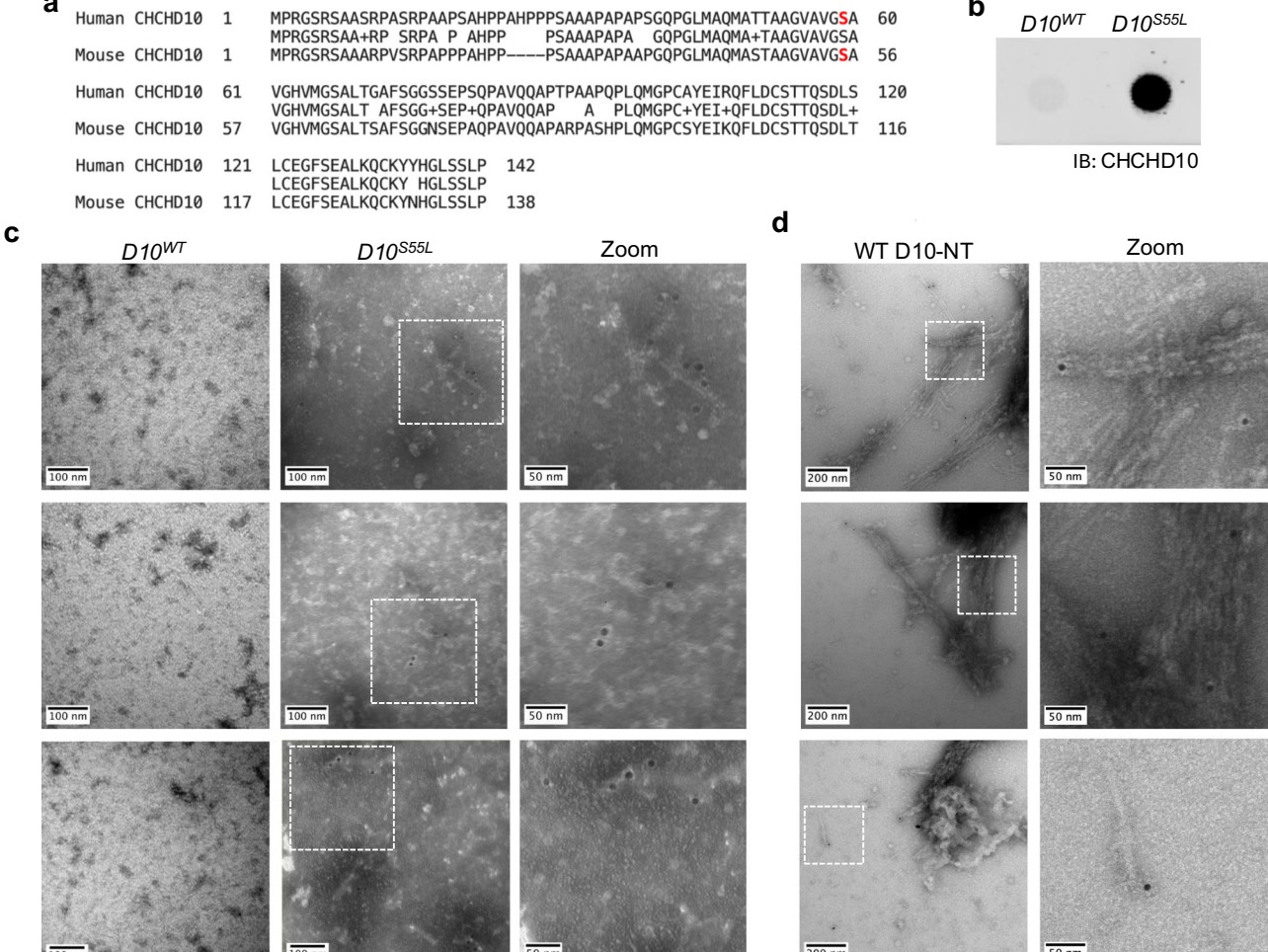

**Fig. 3 | D10 forms amyloid fibrils in a mouse model of disease. a** Sequence alignment of human and mouse D10 showing identical residues (middle), conservative substitutions (+), and non-conservative substitutions (blanks), with residues S59 (human) and S55 (mouse) in red. **b** Filter trap capture of sarcosyl-insoluble extracts from D10$^{WT}$ and D10$^{S55L}$ mouse hearts, immunoblotted (IB) for D10. Similar results were obtained from a second independent sample/experiment. An uncropped image is provided as a Source Data file. **c** TEM micrographs of D10-immunogold labeled sarcosyl-insoluble fraction of heart tissue from D10$^{WT}$ or D10$^{S55L}$ mice. Similar results were obtained from a second independent sample/experiment. **d** TEM micrographs of D10-immunogold labeled in vitro assembled fibrils of WT D10-NT. Similar results were obtained from a second independent sample/experiment.

1, 2.31 Å resolution) revealed an ordered core formed by residues P42-G75 and containing two identical protofilaments (PFs) related by pseudo-2$_1$ helical screw symmetry (Fig. 4a and Supplementary Fig. 2). Each PF contains three beta-strands, extending from residues M45 to A53, V55-V57, and V64-A72. Strand-2 forms part of a beta-helix-like turn structure, commonly observed in amyloid fibrils, connecting the first and third strands. The interface between these PFs is formed primarily between the tip of each beta-helix and the N-terminal half of strand 1. Polymorph-2, solved to 2.70 Å resolution, also contains two identical symmetry-related PFs, with a fold (PF2) distinct from that found in polymorph-1 (PF1) (Fig. 4b and Supplementary Fig. 3). PF2 is comprised of residues G43–G76 and exhibits four strands consisting of residues M45–A53 (similar to strand-1 in PF1), V55–V57 (similar to strand-2 in PF1), A60–V64 and L69–F73. An extensive inter-PF2 interface is comprised of strands 2–4. WT D10-NT polymorph-3, solved to 2.63 Å resolution, contains four identical PFs with yet a different fold (PF3) arranged as two helically symmetric dimers situated around a central pseudo-2-fold helical symmetry axis (Fig. 4c and Supplementary Fig. 4). PF3 is comprised of residues T50–G76 and contains only two beta strands consisting of T51–V57 and A68–V70, with an extensive turn between them. The central interface is quite small, consisting primarily of residues M65 and G66, while the PF interface within each

symmetric dimer is extensive and comprises strand 1 and the first half of the turn between strands 1 and 2. In addition to the three structures we were able to solve, 2D classification of the datasets revealed the likely presence of additional polymorphs that we were not able to resolve, indicating that the full landscape of D10-NT amyloid fibrils may be still more complex, as has been seen with other amyloid-forming proteins[28,29].

We also determined the structure of D2-NT amyloid fibrils to 2.03 Å resolution, which again contains two identical PFs related by a helical symmetry axis (Fig. 4d and Supplementary Fig. 5). The core ordered region of each D2-NT PF consists of D2 residues G54–G86. A sequence alignment of D10 and D2 (Fig. 4e) shows that this region corresponds to residues G43–G75 of D10 and features only 5 sequence differences between the two proteins. Therefore, the core ordered regions of D2-NT and D10-NT fibrils are formed by highly homologous regions within each protein. While the D2-NT PF fold differs from those of all three D10-NT PFs, certain elements are conserved. The D2 PF fold contains three strands consisting of residues L55–T61, V66–V68, and S70–T81. Strand-1 adopts a very similar conformation to that observed in the D10-NT PF1 structure, as do the last five residues of strand-3, and the two strands pack together in a nearly identical manner in both structures, with residue F73(D10)/

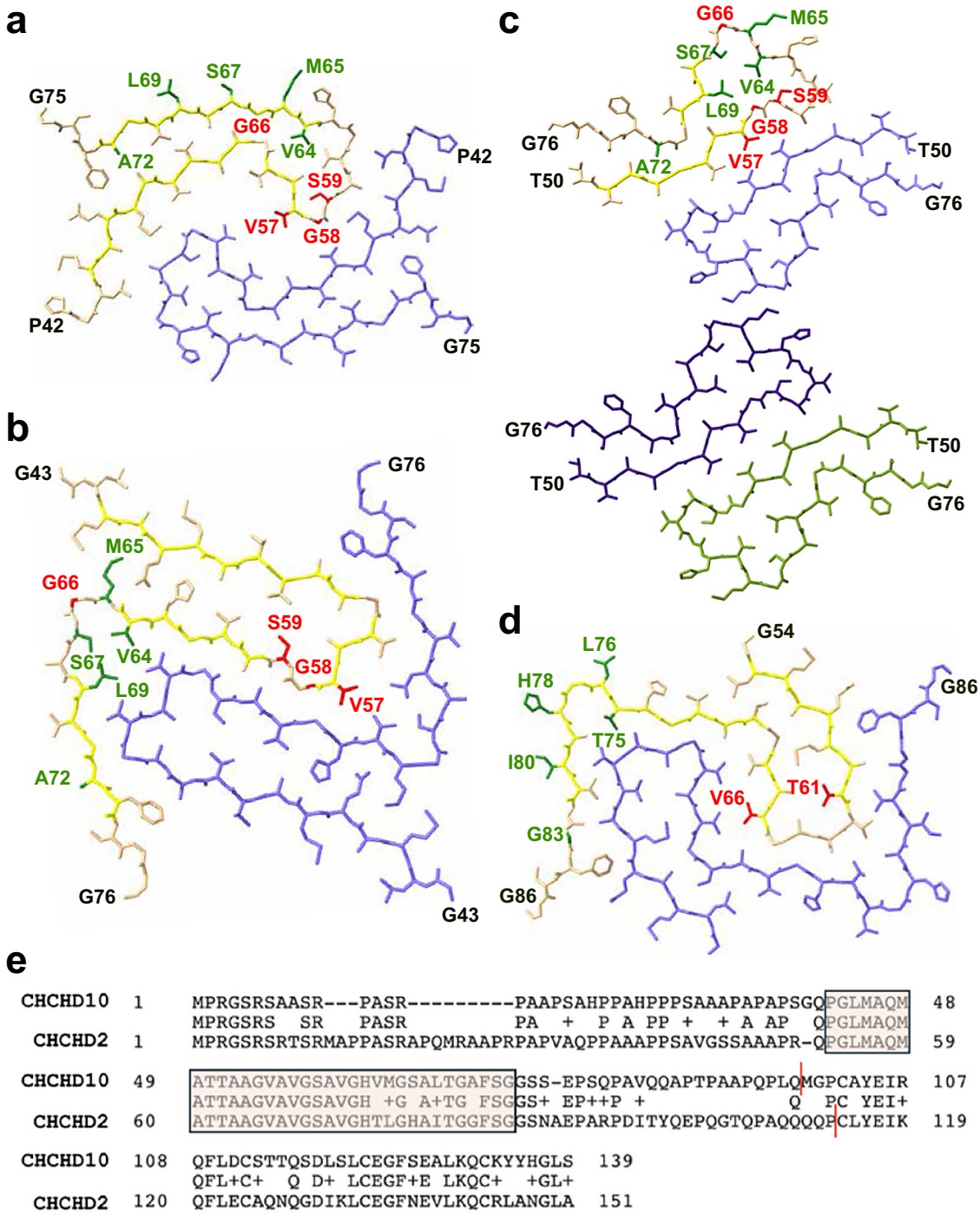

**Fig. 4 | A conserved region forms the structured cores of D10 and D2 amyloid fibrils. a–c** Stick representations of single layers of the cryo-EM structures of the ordered fibril cores of three D10-NT polymorphs. Beta strands are highlighted in yellow. Sites of disease-linked mutations are shown in red, with S59L situated in the interior of all three PF folds, while V57E and G58R are found either in the PF interior or in the PF interface. Amino acids that differ between D10 and D2 are shown in green, with M65, S67, and L69 exposed on the surface of D10-NT polymorph-1. **d** Stick representations of a single layer of the cryo-EM structure of the ordered fibril core of D2-NT fibrils. Beta strands are highlighted in yellow. Sites of disease-linked mutations are shown in red, with T61 situated in the interior of the D2-NT PF folds, while V66 projects into the PF interface. Amino acids that differ between D10 and D2 are shown in green, with L76, H78, and I80 exposed on the D2 fibril surface. **e** Sequence alignment of human D10 (top) and D2 (bottom) showing identical amino acids (middle), conservative substitutions (+), and non-conservative substitutions (blanks). The fibril core is boxed and shaded. Red lines indicate the C-termini of the N-terminal constructs used in our studies.

F84(D2) packing against the equivalent Q47(D10)/Q58(D2) residue (Fig. 5a–d). However, in the D2-NT structure, strand-1 of one PF (PF-A) packs against strand-3 of the other PF (PF-B), because the beta-helix that connects the two strands in the D10-NT structure reverses on itself, allowing the C-terminal region of D2-NT PF-A to occupy the location of the C-terminal region of PF-B in the D10-NT structure. In

addition to this, strand-2 of the D2-NT is superposable on the corresponding VAV sequence of all three D10-NT PFs (Fig. 5e) and forms the central part of the PF interface both in the D2-NT structure and in D10-NT polymorph-1 (Fig. 5a, b).

Notably, the five sequence variations in the core ordered region of D2 compared with D10 occur either at positions facing the outside of

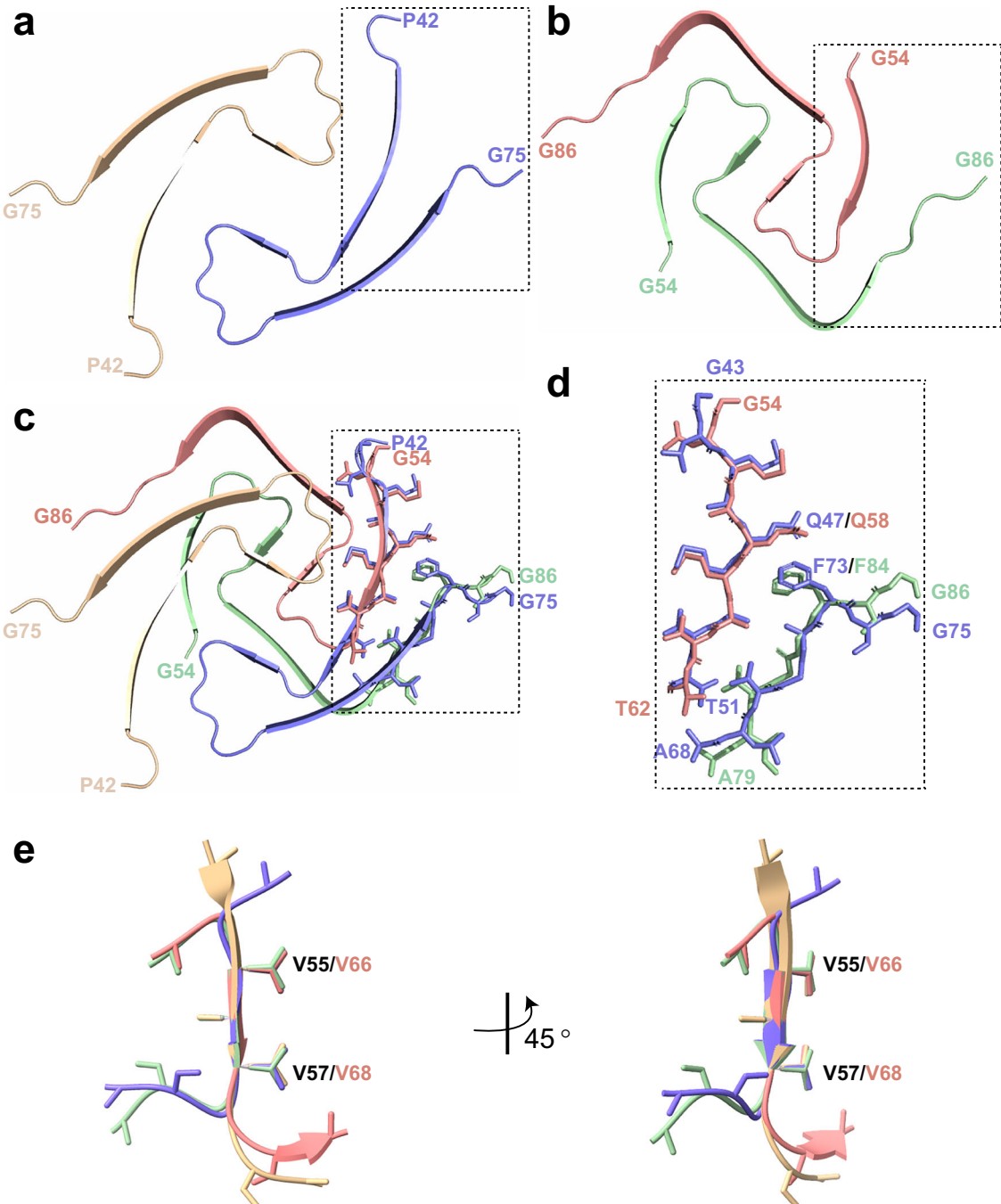

**Fig. 5 | D10 and D2 amyloid fibrils feature conserved structural elements.**
**a, b** Secondary structure of ordered fibril cores of WT D10-NT polymorph-1 (**a**) and
D2-NT (**b**) showing the intra-molecular (D10-NT) vs. inter-molecular (D2-NT)
packing arrangement of strands 1 and 3. Individual protofilaments (PFs) are shown
in tan, blue, salmon, and green. **c, d** Overlay of D10-NT and D2-NT structures,
highlighting the similar conformations of the N-terminus of strand-1 (D10 residues

G43-T51; D2 residues G54-T62) and the C-terminus of strand-3 (D10 residues A68-
G75; D2 residues A79-G86) and of their detailed packing arrangement. **e** Overlay of
the V55-V57 VAV motif from all three WT D10-NT polymorphs with the corre-
sponding V66-V68 VAV motif in the WT D2-NT structure (D10-NT polymorph-1,
purple; D10-NT polymorph-2, pale green; D10-NT polymorph-3, wheat; D2-NT,
salmon).

the D2-NT fibril structure (D2 residues L76, H78 and I80), or are isos-
teric to (T75 in D2 vs. V64 in D10) or smaller than (G83 in D2 vs. A72 in
D10) the corresponding residues in D10 (Fig. 4d). This suggests that
D10 molecules could potentially template onto D2 fibrils. Similarly, the
location of the equivalent residues in the D10-NT PF1 fold (Fig. 4a)
suggests that it could accommodate D2 molecules. However, in the
PF2 and PF3 D10-NT folds, some of the sequence variation sites are
positioned within the PF fold, suggesting that these D10-NT folds may
not be compatible with the D2 sequence.

### Fibril core disease mutations promote alternative D10-NT and D2-NT protofilament folds

A number of disease-associated D10 mutations fall within the core
regions of all three PF structures, including p.V57E, p.G58R, p.S59L,
and p.G66V[3]. Interestingly, residue S59 is a small sidechain that is
packed inside the PF interior in all three wildtype (WT) D10-NT struc-
tures, while residues V57 and G58 are found either in the PF interior or
at PF packing interfaces in all three D10-NT structures (Fig. 4a–c).
Disease-associated substitutions of any of these three sites to the

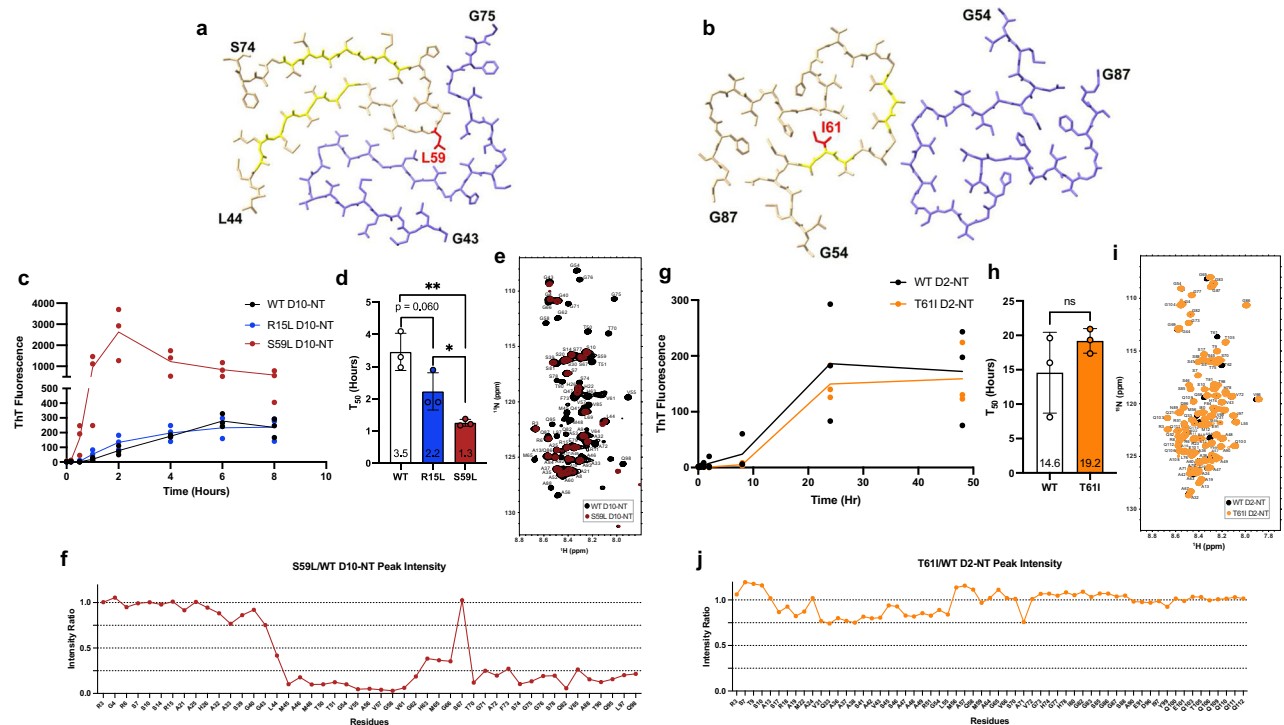

**Fig. 6 | Disease mutations alter the structure of D10 and D2 amyloid fibrils.**
**a** Stick representation of the cryo-EM structures of a single layer of the ordered fibril core of S59L D10-NT fibrils. Residue L59 is shown in red. Beta strands are highlighted in yellow. **b** Stick representation of the cryo-EM structures of a single layer of the ordered fibril core of the T61I D2-NT fibrils. Residue I61 is shown in red. Beta strands are highlighted in yellow. **c** ThT monitored aggregation of purified recombinant WT (black), S59L (red), and R15L (blue) D10-NT. **d** ThT $T_{50}$ for WT, S59L and R15L D10-NT assembly (mean ± SD, $n = 3$ biological replicates, unpaired two-tailed parametric Student's $t$-test; ns $p = 0.060$, * $p = 0.045$, ** $p = 0.003$). **e** Overlaid NMR $^{15}$N-$^1$H HSQC spectra of S59L (red) and WT D10-NT (see

Supplementary Fig. 17 for a full-page version). **f** Ratios of S59L D10-NT NMR resonance intensities to those of corresponding WT D10-NT resonance intensities. **g** ThT monitored aggregation of purified recombinant WT (black) and T61I (orange) D2-NT. Data for WT D2-NT are the same as in Fig. 2a. **h** ThT $T_{50}$ for WT and T61I D2-NT assembly (mean ± SD, $n = 3$ biological replicates, unpaired two-tailed parametric Student's $t$-test; $p = 0.26$. **i** Overlaid NMR $^{15}$N-$^1$H HSQC spectra of T61I (orange) and WT (black) D2-NT (see Supplementary Fig. 18 for a full-page version). **j** Ratios of T61I D2-NT NMR resonance intensities to those of corresponding WT D12-NT resonance intensities. Source data for **c**, **d**, **f**–**h**, **j** are provided as a Source Data file.

larger sidechains associated with disease would be inconsistent with the observed WT structures and would require structural rearrangements. In the case of p.G66V, the PF2 or PF3 structures could potentially accommodate the mutation, but it would not be consistent with the PF1 structure and would therefore be expected to eliminate some of the possible WT PF folds. Thus, disease-associated D10 mutations could bias the D10 fibril landscape in favor of specific, possibly novel, folds. These effects could in turn underlie the different phenotypes associated with each mutation and constitute the basis for structural disease strains, as observed for other amyloids[31,32]. To assess this, we solved the structure of S59L D10-NT fibrils to 3.06 Å resolution (Fig. 6a and Supplementary Fig. 6). As expected, residue L59 in this structure flips away from the tightly packed position of S59 in the PF interior. This results in a different PF fold and packing arrangement. Interestingly, S59L fibrils are no longer symmetric, with each PF adopting a different fold. S59L PF-A adopts a fold similar to the WT PF1, except that the beta-helix connecting strands 1 and 3 is disrupted by the flipping of residue 59. S59L PF-B, however, adopts a completely different fold, likely driven by the fact that the WT arrangement could not be maintained with the bulky L59 at the PF interface.

D2 also features disease-linked mutations in the region constituting the core of D2-NT fibrils. Notably, the most commonly reported pathogenic D2 mutation p.T61I[33] occurs at a position facing the interior of the D2 PF fold (Fig. 4d), suggesting that this mutation, too, could promote alternative fibril structures. To assess this, we solved the structure of T61I D2-NT fibrils to 3.09 Å resolution (Fig. 6b and Supplementary Fig. 7). These fibrils featured a novel PF fold, consisting of residues G54–G87, that differs from the WT D2-NT fold,

all three WT D10-NT folds, and the S59L D10-NT fold. This fold features only two short beta-strands, A60–T62 and G65–V68, the latter encompassing the VAV motif that is present in every fibril form we have observed. In addition, the T61I PF fold appears to contain several LARKS (low complexity aromatic-linked kinked segments)[34], likely stemming from three glycine-aromatic dipeptides (G59–H60, G63–H64, and G69–F70). Unlike the case of S59L, the T61I mutation does not flip the larger isoleucine sidechain to the PF fold exterior. However, the two valines in the VAV motif, which are found at the PF interface in all other protofilament structures we have observed, are flipped towards the PF interior in the T61I fold. The VAV motif in D2 is the site of a mutation reported to be associated with MSA, p.V66M[13], which would also be inconsistent with our WT D2 structure.

Although we observe different fibril structures for the S59L D10-NT and T61I D2-NT mutants, the polymorphism we observe for WT D10-NT makes it difficult to establish whether the observed mutant structures are uniquely associated with the mutants, whether the mutations alter the D10 or D2 polymorphism landscape, or whether the mutant structures are simply additional polymorphs. To examine the reproducibility of the WT and mutant structures, we obtained and analyzed additional data sets for independent preparations of WT D10-NT, S59L D10-NT, and WT D2-NT fibrils (see methods section on Cryo-EM sample preparation and data collection). For WT D10-NT, we obtained 2D classes closely resembling those observed for WT D10-NT polymorph-2 (Supplementary Fig. 8). For S59L D10-NT, we obtained two 2D classes that closely resemble those observed in the sample that led to the S59L D10-NT structure (Supplementary Fig. 9), with similar populations. For WT D2-NT, we again obtained 2D classes highly

similar to those observed in the original sample from which we solved the structure (Supplementary Fig. 10).

## Disease-associated mutations exhibit differential effects on aggregation kinetics

Disease-associated mutations of amyloid proteins can enhance aggregation rates[35,36]. We examined the effects of mutations in D10 and D2, for which we obtained fibril structures, on the aggregation kinetics of D10-NT and D2-NT. The S59L mutation resulted in greatly enhanced D10-NT aggregation rates, as measured by ThT fluorescence (Fig. 6c, d). Furthermore, NMR spectra of this mutant prepared freshly in solution exhibited dramatic signal loss in the region corresponding to the fibril core (Fig. 6e, f), suggesting that intermolecular interactions are greatly enhanced for this mutant, consistent with its faster aggregation rates. In contrast, the aggregation rates of the T61I D2 mutant did not result in substantial alteration of the aggregation kinetics of D2-NT or appreciable signal loss in NMR spectra (Fig. 6g–j). Thus, as for other amyloids[37,38], disease-associated mutations in D10 and D2 exert variable effects on in vitro aggregation kinetics.

Although many mutations associated with neurodegeneration are situated within the ordered cores of D10-NT and D2-NT fibrils, other disease-associated mutations occur outside these regions, including the p.R15L mutation in D10, for which we observed evidence of amyloid formation in patient fibroblasts (Fig. 2e). We solved the structure of R15L D10-NT fibrils to 3.15 Å resolution to assess whether this mutation alters fibril structure despite being outside the fibril core. The resulting structure (Supplementary Fig. 11) was nearly identical to the WT D10-NT polymorph-2 structure, indicating that this mutation likely does not alter D10 fibril structure. Interestingly, however, this mutation appeared to enhance D10-NT aggregation, as assessed via ThT fluorescence (Fig. 6c, d), although not nearly to the extent of the S59L mutation. This mutation also leads to signal loss in NMR spectra (Supplementary Fig. 12a, b) comparable to that observed for S59L, suggesting that it too promotes intermolecular interactions. In the case of R15L and S59L D10, the introduction of a large hydrophobic residue is likely responsible for enhanced intermolecular interactions preceding aggregation. However, the T61I mutation in D2, which also incorporates a large hydrophobic residue, did not result in substantial changes to the aggregation kinetics of the protein or in appreciable signal loss in NMR spectra, indicating that other factors beyond hydrophobicity influence aggregation kinetics.

## Discussion

Here we show for the first time that D10 and D2 form amyloid fibrils in vitro and, for D10, in mouse models of disease and in cells derived from human patients. Structures of D10 and D2 fibrils assembled in vitro from the low complexity N-terminal domains of each protein show that their core comprises the most highly conserved region between the two proteins. In vitro assembled D10-NT fibrils exhibit a range of polymorphs, as has been observed for other amyloid proteins associated with disease[28,29]. At present, we do not have structures of ex vivo D10 or D2 fibrils, but despite potential differences, specific features of ex vivo fibrils are often captured by in vitro fibril structures[28,39]. For D10 and D2, the central conserved VAV motif forms a beta-strand that is associated with the protofilament interface in all the structures we have observed, suggesting that it may be a critical element in fibril formation and may be expected to appear in ex vivo structures as well. This motif also contains disease-associated mutations, p.V57E in D10 and p.V66M in D2, and is immediately proximal to two additional D10 disease-linked mutations, p.G58R and p.S59L, providing further evidence for its importance. Each of these mutations replaces a smaller side chain with a larger one that is incompatible with the observed WT structures and would be expected to result in structural changes. Indeed, the p.S59L D10 mutation and the p.T61I D2 mutation promote profound changes to fibril structure, including

breaking the symmetry observed in WT D10 structures and promoting alternative PF folds.

The effects of disease-linked mutations situated within the fibril core on the resulting fibril structures support the potential relevance of D10 and D2 fibril formation to disease etiology and raise the possibility that these mutations may generate distinct structural strains of D10 and D2 pathology and disease, as has been observed for other amyloids[40]. Supporting this possibility is the wide range of disease presentations and phenotypes reported for different D10 and D2 mutations[2]. However, our observation of multiple WT D10-NT structures suggests the possibility that WT D10-NT and D2-NT could also adopt the structures we observed for the S59L and T61I mutants. We have provided evidence in the form of 2D classes obtained from cryo-EM datasets of independent samples that the WT D10-NT polymorph-2, the S59L D10-NT and the WT D2-NT structures we determined are reproducible and may therefore be preferred by their respective variants. However, resolving the structural effects of mutants unambiguously must await a comprehensive exploration of the polymorph landscape of the WT and mutant proteins. Notably, even if mutations do not lead to unique fibril structures, they may favor or promote alternative structures compared to the WT proteins, both in vitro and in patients.

Disease-linked mutations are also found outside the fibril core regions of D10 and D2. Other known amyloids also feature disease-linked mutations outside their fibril core regions (e.g. p.R5H for tau[41], p.A30P for alpha-synuclein[42]). Such mutations are inherently less likely to influence fibril structure, but may still alter the fibril assembly process. We show here that the p.R15L D10 mutation, situated far outside the D10-NT fibril core region, adopts the polymorph-2 structure of WT D10-NT, providing further evidence, albeit from a single sample and structure, that this fold is favored when the WT D10-NT sequence is preserved in the fibril core region.

It is notable that we observed fibril formation for WT D10 and D2, suggesting that mutation of these proteins may not be required for their aggregation. Disease-associated aggregation of WT proteins is commonly observed for other amyloids, for example for alpha-synuclein[43,44], tau[45,46] and TDP-43[47,48]. Indeed, WT D2 is reported to co-aggregate with its mutated counterparts[33], and expression of WT human D2 in flies leads to motor impairment[49]. We also observe evidence for WT D10 aggregation in filter trap blots of very old (P500+) mice (Supplementary Fig. 13). At present, there are no published reports associating D10 and D2 aggregation with disease in the absence of D10 or D2 mutations, and aggregation of the WT protein at advanced ages may occur without disease, as has been observed for Amyloid beta[50] or TMEM106B[51]. Nevertheless, our results suggest that such an association is possible and should be investigated.

The core fibril regions of D10 and D2 are highly conserved, featuring only 5 sequence variations. At least one of the D10 fibril polymorphs we observe could likely accommodate these variations and also features structural elements that are conserved in our D2 fibril structure. These observations suggest the possibility that D10 and D2 form fibrils containing both proteins, especially as the two proteins are reported to form heterodimers[52] and to co-aggregate in vivo[5]. However, our attempts to seed D2-NT fibrils with preassembled D10-NT fibrils did not result in a shorter D2 assembly lag phase, suggesting a lack of cross-templating under our conditions. Notably, mixed amyloid fibrils have only rarely been observed[53], despite the fact that other amyloids like tau feature homologous isoforms that populate polymorphs with structural similarities. Further work will be required to more thoroughly assess whether D10 and D2 can form fibrils with each other, or possibly with non-homologous associated proteins like alpha-synuclein[54], as has recently been reported for TDP-43 and annexin-A11[55].

Another important outstanding question to be addressed in future studies is the mechanisms whereby fibrillar D10/D2 aggregates

may participate in mitochondrial stress and disease pathogenesis. It is possible to speculate that fibrils interfere with mitochondrial membrane structures, especially the inner mitochondrial membrane, which is very densely packed with protein complexes that ensure cristae maintenance, and which is key for respiratory chain and ATPase assembly and function as well as numerous critical metabolite and protein transport systems.

## Methods

### Ethical statement
All research described here complies with all relevant ethical regulations and was approved by the Weill Cornell Medicine Institutional Animal Care and Use Committee (under protocol number 0610-549 A) for mouse studies or by the Columbia University Irving Medical Center's Institutional Review Board and the Human Embryonic and Human Pluripotent Stem Cell Research Committee (under protocol number IRB-AAAK2000 (Y10M01)) for human patients. All animal procedures were conducted in accordance with Weill Cornell Medicine (WCM) Animal Care and Use Committee and performed according to the Guidelines for the Care and Use of Laboratory Animals of the National Institutes of Health. Written informed consent for the collection of skin biopsy samples was obtained under a protocol approved by Columbia University Irving Medical Center's Institutional Review Board and the Human Embryonic and Human Pluripotent Stem Cell Research Committee. This protocol allows for the use of clinical data and de-identified demographic information for research purposes, as contained in this publication.

### Protein expression and purification
Plasmids encoding human CHCHD10, CHCHD2, the CHCHD10 N-Terminus (residues 1-98) and the CHCHD2 N-Terminus (residues 1-113) preceded by an N-terminal 6x-His-SUMO tag were procured from Twist Biosciences. The p.S59L and p.R15L CHCHD10 mutants and the p.T61I D2 mutant were generated using an In-Fusion Cloning kit (Takara Bio) and confirmed by DNA sequencing (Genewiz). Recombinant proteins were expressed in *E. coli* BL21/DE3 cells (Novagen) grown in either LB Broth or M9 minimal media containing [15]N-labeled ammonium chloride (1 g/L) or [15]N-labeled ammonium chloride and [13]C-labeled D-glucose (2 g/L) at 37 °C (275 rpm) induced with 1 mM IPTG (Isopropyl b-D-1-thiocalactopyranoside) at OD 600 nm of 0.6–0.8. Four hours post induction, cells were harvested via centrifugation at ca. 10,500 × g at 4 °C for 15 min. Cell pellets (stored at -20 °C overnight) were resuspended in 50 mL lysis buffer (20 mM Tris (pH 8.0), 350 mM NaCl, 20 mM Imidazole, 1 mM PMSF (phenylmethylsulfonyl fluoride), 1 mM EDTA and 3 mM bME (β-mercaptoethanol)) and lysed using an EmulsiFlex-C3 (AVESTIN, Ontario, Canada), followed by centrifugation at ca. 40,000 g for 1 h to remove cellular debris. The supernatant was loaded onto a Ni-NTA column equilibrated using 20 mM Tris (pH 8.0), 350 mM NaCl, 20 mM Imidazole, 3 mM bME, washed with the same buffer, and the SUMO-tagged protein was eluted using 20 mM Tris (pH 8.0), 350 mM NaCl, 250 mM Imidazole, 3 mM bMe. Protein-containing fractions were pooled and cleaved overnight at 4 °C using SUMO protease (added to final concentration ca. 1 μM), followed by dialysis against 20 mM Tris (pH 8.0), 150 mM NaCl, and 1 mM DTT and loaded again onto a Ni-NTA column. The cleaved N-terminal constructs were collected in the flowthrough, loaded onto a 5 mL HiTrapTM SP HP column on an AKTA Pure Protein Purification System (GE) and eluted with 20 mM Tris (pH 8.0) 1 M NaCl. Purified constructs were then exchanged into deionized H2O using a PD-10 Column (Cytiva, Marlborough, MA) and lyophilized.

### Circular dichroism spectroscopy
CD measurements were performed on an AVIV 410 CD spectropolarimeter. Spectra were obtained from 300-190 nm at 25 °C after a two-minute temperature equilibration with a wavelength step of 1 nm,

an averaging time of 5 s, 1 scan per sample and a cell pathlength of 0.02 cm (Starna, Atascadero, CA). Backgrounds were collected and subtracted from all spectra. Final construct concentrations ranged from ca. 50–100 μM as assessed by 1D proton NMR with DSS as an internal standard. The dearth of aromatic residues in all the polypeptides used made reliable determination of absolute protein concentrations exceedingly difficult. Therefore, CD data are presented in millidegrees and were not converted to mean residue molar ellipticity.

### Thioflavin T aggregation reactions
In vitro WT and mutant CHCHD10 and CHCHD2 aggregation was monitored using a ThT protocol adopted from a report on tau aggregation[56]. Lyophilized protein was brought up to stock concentrations of ca. 16 μM in aggregation assay buffer (20 mM Tris, pH 7.4, 100 mM NaCl, 1 mM EDTA and 1 mM DTT) and filtered using 100 kDa centrifugal filter (AMICON) at room temperature for 10 mins at ca. 15,000 × g to remove any initial aggregate formation and diluted 2-fold with aggregation assay buffer to a final concentration of 8 μM. A 3 mM ThT stock was prepared in aggregation assay buffer and filtered using a 0.2 μm filter (Pall). Aggregation was induced using an Eppendorf Thermomixer R at 37 °C, shaking at 1000 or 1200 rpm. Data were collected in triplicate at designated time intervals using a microplate fluorescence reader (Molecular Devices). The excitation and emission wavelengths were 450 nm and 510 nm, respectively. Time to reach half maximal ThT fluorescence ($T_{50}$) was determined for each sample using a scatter plot of ThT fluorescence over time with smoothed lines in Excel (Microsoft). Decreased values observed at later time points in some of the curves may be due to the formation of larger species that settle out of the excitation volume. Unpaired Student's t-tests were performed in Prism (GraphPad) with a confidence levels set to 95% ($p < 0.05$).

### Animals
Chchd10S55L knock-in mice were previously generated (C57BL6 strain) by CRISPR/Cas9 approach[5] and are available as Stock #028952 from the Jackson Laboratory. Breeding was set up between Chchd10S55L heterozygous (Het) males with wildtype (WT) C57BL/6NJ females (Jackson Laboratory, stock #005304). Tail DNA was extracted with a Promega kit and genotype was determined by sequencing services at Transnetyx.

### Isolation of mitochondria
Mitochondria were freshly isolated from brain and spinal cord as previously described[57] with minor adjustments. Briefly, whole mouse hearts were minced on ice, washed, and incubated in 0.01% Trypsin-EDTA (Invitrogen) in PBS (ThermoScientific Scientific) on ice for 30 min. Tissue was then rinsed and homogenized 40 times using a small glass homogenizer with MS-EGTA buffer (225 mM D-mannitol, 75 mM sucrose, 20 mM Hepes, 1 mM EGTA, 1 mg/mL fatty-acid-free BSA, pH 7.4). Homogenates were subjected to differential centrifugation to obtain fractions containing intact mitochondria, used for filter trap assays. Mitochondrial proteins were quantified using the Bradford protein assay (Bio-Rad).

### Extraction of sarcosyl-insoluble fractions
Sarcosyl-insoluble fractions were obtained as previously described[58] with minor adjustments. 20 mg of heart tissue from one female D10[WT] mouse and one female D10[S55L] mouse (age post-natal day 340 (P340)) was homogenized in 1 ml of extraction buffer (10 mM Tris-HCl, pH 7.4, 0.8 M NaCl, 10% sucrose, 1 mM EGTA, 2% sarcosyl, 1× cOmplete protease inhibitor) using a Tissue-Tearor (BioSpec) for 1 min on ice. Homogenized tissue was incubated at 37 °C for 1 h, then centrifuged at 10,000 × g for 10 min at 4 °C to remove debris. Supernatant was collected and spun at 100,000 × g for 1 hour at 4 °C. Pellet was resuspended in 150 μl extraction buffer and centrifuged at 3000 × g for

5 min at 4 °C. Supernatant was raised to 1 ml in 50 mM Tris-HCl, pH 7.4, 150 mM NaCl, 10% sucrose, 0.2% sarcosyl, and 1× cOmplete protease inhibitor, then centrifuged at 100,000 × *g* for 30 min at 4 °C. Sarcosyl-insoluble pellet was resuspended in 50 μl 20 mM Tris-HCl, pH 7.4, 50 mM NaCl and was used for filter trap assay, immunogold-labelling, and electron microscopy. Of note, N-Lauroylsarcosine (sarcosyl) sodium salt (Sigma) was used for these experiments.

### Filter trap assay

Insoluble protein aggregates were detected by filter trap assay as previously described[59]. Briefly, 5 μg of mitochondria incubated with 1% NP-40 for 30 min on ice or 1 μL of the final sarcosyl-insoluble fraction were loaded onto a Bio-Dot Microfiltration apparatus (Bio-Rad) containing a cellulose acetate membrane (0.2 μm pore diameter, Whatman). Vacuum was applied to pass samples through the membrane, which was then washed with 1% Tween-20 in PBS. Trapped proteins were detected with rabbit anti-CHCHD10 (Abcam Ab121196) and anti-rabbit HRP. Blots were then imaged using Clarity Western ECL Blotting Substrates (Bio-Rad) and imaged on ChemiDoc Touch (Bio-Rad).

### Immunogold preparation of sarcosyl-insoluble fractions for EM

Five microliters of sarcosyl-insoluble sample was applied to a formvar-carbon coated grid and allowed to settle for 2 min. Grids were inverted onto a 100 μl drop of Aurion Blocking buffer for the secondary antibodies (Aurion, Electron Microscopy Sciences) for 15 min. Grids were incubated on 100 μl drops of primary rabbit anti-CHCHD10 (Abcam Ab121196) in PBS-c (PBS + 0.1% BSA-c, Aurion) for 1 h in a humid box at room temperature. Grids were washed 5 times in PBS-c for 5 min. Grids were incubated on 100 μl drops of gold-tagged secondary antibody (Aurion) for 1 hour in a humid box, then washed 3 times for 3 min in PBS-c and two times for 3 min in deionized H$_2$O. Grids were fixed with 2.5% buffered glutaraldehyde (Sigma) and washed 3 times for 1 min in deionized H$_2$O. Negative stain was applied with 1.5% (aq) uranyl acetate (Sigma), blotted, and dried. EM imaging used a JEOL JEM 1400 transmission electron microscope operated at 100 kV and equipped with an Olympus-SIS Veleta side-mount 2 K × 2 K digital camera.

### Fibroblast imaging

Fibroblasts from a male patient, age 60, with D10 p.R15L mutation (ALS diagnosis and the variant CHCHD10 c.44 G > T) and a female control, age 55, were cultured on glass coverslips (Electron Microscopy Sciences 50-949-008) in DMEM with 10% FBS to 80% confluency before fixation with 4% paraformaldehyde. Fixed fibroblasts were permeabilized with 0.1% Triton X-100 in PBS, blocked, and then stained with anti-CHCHD10 primary antibody (ProteinTech 25671-1-AP, Rabbit polyclonal, 1:500 dilution) and ThioS (Sigma-Aldrich T1892) to detect amyloid deposits. ThioS was prepared by dissolving it in 50% ethanol to 0.05% w/v and filtering the solution through a 0.2 μM PES filter (Thermo Scientific 725-2520). For ThioS staining, fibroblasts were incubated with this solution for 5 min at room temperature, followed by sequential washes in 50%, 80% and 95% ethanol then washed several more times in deionized H$_2$O. Fibroblasts were then imaged by laser scanning confocal microscopy (85 μM pinhole) using a Leica SP5 system equipped with a 40× oil objective. The ThioS staining was captured using 488 nm excitation and 500–538 nm PMT detection. Images were imported into ImageJ2 (Fiji distribution) and the color of the default lookup table (LUT) was changed to create the D10 and ThioS overlays.

### Negative stain electron microscopy

Five microliters of aggregated WT D10-NT or WT D2-NT sample was applied to a formvar-carbon coated grid and allowed to settle for 1 min. Grids were washed with diH$_2$O 3 times, stained with 1.5% (aq) uranyl acetate (Sigma), incubated with stain for 1 min, washed with diH2O, blotted with filter paper and dried. EM imaging used a JEOL JEM 1400 transmission electron microscope operated at 100 kV and equipped with an Olympus-SIS Veleta side-mount 2 K × 2 K digital camera.

### Solution state nuclear magnetic resonance (NMR) spectroscopy

D10-NT and D2-NT constructs were prepared in NMR Buffer (100 mM NaCl, 10 mM Na$_2$HPO$_4$, pH 6.8) at concentrations ranging from ca. 50–150 μM in 5 mm NMR tubes. Relative protein concentrations were corroborated by 1D proton NMR using 4,4-dimethyl-4-silapentane-1-sulfonic acid (DSS) as an internal standard. $^1$H-$^{15}$N HSQC spectra were collected on a Bruker AVANCE 500-MHz spectrometer equipped with a Bruker TCI cryoprobe and acquisition software Bruker Topspin 3.6.5 at 10 °C with 1024 complex points in the $^1$H dimension and 214 complex points in the $^{15}$N dimension using spectral widths of 18 PPM ($^1$H) and 30 PPM ($^{15}$N). NMR spectra were processed using NMRpipe and analyzed using NMRFAM-sparky 3.115 and NMRbox. Backbone resonance assignments for WT D10-NT and D2-NT were made using standard triple resonance experiments (HNCO, HNCA, HNCACB, CBCACONH and HNCACO with 14 PPM spectral width and 1024 complex points ($^1$H), 30 PPM spectral width and 72 complex points ($^{15}$N), and 12/128, 32/140, 70/150, 70/140, 12/128 PPM/complex points ($^{13}$C) respectively) collected on a Bruker Avance III 800-MHz spectrometer equipped with a Bruker TXO cryoprobe and acquisition software Bruker Topspin 3.5.7. For WT D10-NT 75/78, 76/78 and 230/281 $^1$H, $^{15}$N and $^{13}$C backbone and C$_\beta$ resonances were assigned and for WT D2-NT, 93/93, 93/93 and 305/321 $^1$H, $^{15}$N and $^{13}$C backbone and C$_\beta$ resonances were assigned. Assignments for mutants were transferred from the WT assignments by inspection. Intensity ratios were normalized by the average ratio of the four N-terminal residues to account for any differences in sample concentrations.

### Cryo-EM sample preparation and data collection

Fibrilization of WT D10-NT, S59L D10-NT, R15L D10-NT, WT D2-NT, or T61I D2-NT was performed as described in Table 1, monitored using ThT fluorescence. Aliquots (3.5 μL) of each sample were applied to a glow-discharged holey copper grid (Quantifoil R1.2/1.3, 300 mesh) and incubated for 30 s. The grids were blotted for 3 s under 100% humidity at 20 °C and plunge-frozen into liquid ethane using a Vitrobot Mark IV (FEI, Thermo). Data sets for WT D10-NT sample 1 (polymorph-1 and -3), S59L D10-NT and R15L D10-NT were collected using Leginon[60] at the

### Table 1 | Summary of cryo-EM samples

| Sample | Concentration (μM) | Shaking speed (rpm) | Temp (°C) | Time point (hrs) | Polymorphs with twist[a] | Solved structures |
|---|---|---|---|---|---|---|
| D10-NT sample-1 (polymorph1/3) | 80 | 1200 | 37 | 120 | 2 | 2 |
| D10-NT sample-2 (polymorph-2) | 80 | 1200 | 37 | 120 | 1 | 1 |
| D2-NT | 80 | 1200 | 37 | 192 | 2 | 1 |
| S59L D10-NT | 80 | 900 | 37 | 22 | 2 | 1 |
| T61I D2-NT | 80 | 1000 | 37 | 24 | 1 | 1 |
| R15L D10-NT | 80 | 1000 | 37 | 2 | 1 | 1 |

[a]The number of polymorph classes featuring twisted fibrils was assessed using the FilamentTools procedure in RELION 5.0b.

**Table 2 | Statistics of cryo-EM data collection and refinement**

|  | D1O-NT-PM1 | D1O-NT-PM2 | D1O-NT-PM3 | D2-NT |
|---|---|---|---|---|
| PDB ID | 9CWW | 9OYQ | 9OYW | 9OYR |
| EMDB ID | EMD-45976 | EMD-71031 | EMD-71037 | EMD-71032 |
| | | Data collection | | |
| Magnification | 64,000 | 100,000 | 64,000 | 105,000 |
| Pixel size (Å) | 1.076 | 1.16 | 1.076 | 0.825 |
| Defocus Range (µm) | 0.9 to 2.2 | 0.8 to 2.5 | 0.9 to 2.2 | 0.6 to 2.5 |
| Voltage (kV) | 300 | 200 | 300 | 300 |
| Energy filter | 20 eV | 20 eV | 20 eV | 20 eV |
| Microscope/camera | Krios/K3 | Talos Glacios /Falcon 4i | Krios/K3 | Krios/K3 |
| Exposure time (s/frame) | 0.05 | 0.127 | 0.05 | 0.045 |
| Number of frames | 50 | 50 | 50 | 40 |
| Total dose ($e^-$/Å$^2$) | 51.82 | 50 | 51.82 | 48.4 |
| | | Reconstruction | | |
| Micrographs | 8278 | 12,002 | 8278 | 5892 |
| Manually picked fibrils | 5167 | 2459 | 5167 | 5435 |
| Segments extracted (no.) | 1,902,596 | 415,983 | 1,808,471 | 3,593,244 |
| Segments after Class2D (no.) | 667,405 | 351,565 | 47,048 | 63,528 |
| Segments after Class3D (no.) | 602,449 | 351,565 | 29,181 | 23,351 |
| Crossover used for initial models (Å) | 360 | 680 | 720 | 600 |
| Box size for initial model (pixel)[a] | 384 | 512 | 640 | 896 |
| Box size for refinement (pixel)[a] | 256 | 256 | 256 | 256 |
| Resolution (Å) | 2.31 | 2.70 | 2.63 | 2.03 |
| Map sharpening B-factor (Å$^2$) | −56.6379 | −70.2058 | −43.8268 | −23.67 |
| Symmetry | pseudo-$2_1$ | pseudo-$2_1$ | C1 | C1 |
| Helical rise (Å) | 2.365 | 2.38 | 4.73 | 4.796 |
| Helical twist (°) | 178.845 | 179.418 | 1.27 | 1.34 |
| | | Atomic model | | |
| Non-hydrogen atoms | 1272 | 3762 | 5832 | 1664 |
| Protein residues | 204 | 612 | 972 | 264 |
| Ligands | 0 | 0 | 0 | 0 |
| r.m.s.d. Bond lengths | 0.002 | 0.002 | 0.002 | 0.001 |
| r.m.s.d. Bond angles | 0.002 | 0.501 | 0.535 | 0.416 |
| MolProbity score | 0.825 | 1.27 | 1.88 | 1.48 |
| All-atom clashscore | 1.30 | 5.06 | 12.33 | 9.08 |
| Rotamer outliers (%) | 5.49 | 0 | 0 | 0 |
| Ramachandran Outliers (%) | 0 | 0 | 0 | 0 |
| Ramachandran Allowed (%) | 0 | 0 | 4.00 | 0 |
| Ramachandran Favored (%) | 100 | 100 | 96.00 | 100 |
|  | **S59L D1O-NT** | **T61I D2-NT** | **R15L D1O-NT** | |
| PDB ID | 9OYS | 9OYT | 9OYO | |
| EMDB ID | EMD-71033 | EMD-71034 | EMD-71028 | |
| | Data collection | | | |
| Magnification | 64,000 | 100,000 | 81,000 | |
| Pixel size (Å) | 1.076 | 1.16 | 1.083 | |
| Defocus Range (µm) | 0.8 to 2.5 | 0.8 to 2.5 | 0.5 to 2.5 | |
| Voltage (kV) | 300 | 200 | 300 | |
| Energy filter | 20 eV | 20 eV | 20 eV | |
| Microscope/camera | Krios/K3 | Talos Glacios/Falcon 4i | Krios/K3 | |
| Exposure time (s/frame) | 0.05 | 0.125 | 0.05 | |
| Number of frames | 46 | 40 | 40 | |
| Total dose ($e^-$/Å$^2$) | 59.653 | 40 | 51.21 | |
| | Reconstruction | | | |
| Micrographs | 4228 | 8291 | 11,144 | |
| Manually picked fibrils | 1543 | 2459 | 2684 | |
| Segments extracted (no.) | 658,614 | 1,133,246 | 1,931,884 | |

**Table 2 (continued) | Statistics of cryo-EM data collection and refinement**

| | S59L D10-NT | T61I D2-NT | R15L D10-NT |
|---|---|---|---|
| Segments after Class2D (no.) | 61,902 | 315,652 | 152,213 |
| Segments after Class3D (no.) | 19,650 | 38,570 | 13,286 |
| Crossover used for initial models (Å) | 225 | 430 | 720 |
| Box size for initial models (pixel)[a] | 320 | 448 | 512 |
| Box size for refinement (pixel)[a] | 256 | 256 | 256 |
| Resolution (Å) | 3.06 | 3.09 | 3.15 |
| Map sharpening B-factor (Å$^2$) | −50.3765 | −62.64 | −64.4635 |
| Symmetry | C1 | pseudo-$2_1$ | pseudo-$2_1$ |
| Helical rise (Å) | 4.65 | 2.36 | 2.425 |
| Helical twist (°) | 3.88 | 179.04 | 179.3892 |
| **Atomic model** | | | |
| Non-hydrogen atoms | 1218 | 3834 | 3762 |
| Protein residues | 192 | 612 | 612 |
| Ligands | 0 | 0 | 0 |
| r.m.s.d. Bond lengths | 0.002 | 0.001 | 0.001 |
| r.m.s.d. Bond angles | 0.501 | 0.387 | 0.383 |
| MolProbity score | 1.42 | 1.82 | 0.97 |
| All-atom clashscore | 7.7 | 7.45 | 2.00 |
| Rotamer outliers (%) | 0 | 0 | 0 |
| Ramachandran Outliers (%) | 0 | 0 | 0 |
| Ramachandran Allowed (%) | 1.67 | 6.25 | 0 |
| Ramachandran Favored (%) | 98.33 | 93.75 | 100 |

[a]Pixels were not binned.

New York Structural Biology Center. The data set for WT D2-NT fibrils was collected using Leginon[60] at New York University Langone Health's Cryo-EM Laboratory. Data sets for WT D10-NT sample 2 (polymorph-2) and T61I D2-NT were collected using smart EPU at Weill Cornell Medicine's Cryo-EM Core Facility. The detailed parameters of cryo-EM data collection are provided in Table 2. Additional independent samples (independent protein preparations and aggregation reactions) of WT D10-NT, S59L D10-NT, and WT D2-NT were prepared as described in Supplementary Table 1 in order to evaluate reproducibility. The repeat data set of WT D10-NT was collected using smart EPU at Weill Cornell Medicine's Cryo-EM Core Facility, and repeat datasets of S59L D10-NT and WT D2-NT were collected using Leginon[60] at the New York Structural Biology Center. The detailed parameters of the above three cryo-EM data collections to evaluate reproducibility are described in Supplementary Table 2.

### Cryo-EM image processing and helical reconstruction

All datasets were processed using RELION 4.0.0 or 5.0b[61,62]. Frame stacks were aligned with dose-weighting using MotionCorr2[63], and the contrast transfer function (CTF) was estimated using CTFFIND4.1[64]. Helical reconstruction was performed in RELION 4.0.0 or 5.0b[61,62,65]. Fibrils were manually picked and segments were extracted using a large box size, typically 512 pixels, for initial classification. Pixel sizes were 1.076 for WT D10-NT sample 1 (polymorph-1 and -3) and S59L D10-NT, 1.083 for R15L D10-NT, 0.825 for WT D2-NT, and 1.16 for WT D10-NT sample 2 (polymorph-2) and T61I D2-NT (Table 2). Reference-free 2D classification, typically using 100 classes, was conducted for at least 25 iterations to assess the presence of different polymorphs. Polymorphism was further confirmed using the FilamentTools program incorporated in RELION 5.0b[30]. Crossover distance was estimated from the 2D classes using ImageJ. Further re-extraction and 2D classification for selected segments was conducted and the resulting 2D classes were used to generate initial models (Supplementary Fig. 14) using the relion_helix_inimodel2d program[61]. Crossover distances and box sizes of the 2D classes used for initial models are listed in Table 2.

Further re-extraction to a smaller box size (256 pixels) was conducted for selected segments followed by 3D auto-refinement without a mask. A mask was then created and a 3D auto-refinement was conducted with mask. A further 3D auto-refinement was conducted with optimization of helical twist and rise. C1 with pseudo-$2_1$ screw symmetry and C2 symmetry were also separately imposed in the refinements to test if there was further improvement of the maps. Refined unfiltered half-maps were postprocessed using a soft mask spanning 20% of the box along the helical axis. The particles were then Bayesian polished and CTF refined and subjected to a further round of 3D auto-refinement to improve the resolution. 3D classification and another round of 3D auto-refinement was performed to the best class. The final reconstruction was sharpened by applying the standard post-processing procedure. All reconstruction details are shown in Table 2.

The independent datasets of WT D10-NT, S59L D10-NT, and WT D2-NT for evaluation of reproducibility were processed using the same methods as described above. Briefly, fibrils were manually picked and segments were extracted using a box size of 512 and 768 pixels (pixel size 1.16 Å) for WT D10-NT, 384 and 784 pixels (pixel size 1.076 Å) for S59L D10-NT, and 512 and 700 pixels (pixel size 1.076 Å) for WT D2-NT. Reference-free 2D classification, typically using 100 classes, was conducted for at least 25 iterations. Crossover distance was estimated from the 2D classes using ImageJ. Further details of 2D classification are provided in Supplementary Table 2.

### Cryo-EM model building and refinement

Atomic models were built de novo and modified by COOT[66]. Models were iteratively refined using the Real-space refinement package in PHENIX[67] and validated using MolProbity[68]. For cross validation, all atoms in the refined models were randomly displaced by an average of 0.3 Å, and each resulting model was refined against the first unfiltered

half-map obtained from processing (FSC-work). FSC curves were then calculated between the FSC-work model and the second unfiltered half-map obtained from processing (FSC-free), and between the refined models and the full-map (FSC-sum). The structural figures were prepared in UCSF ChimeraX[69] and PyMOL (DeLano Scientific). The detailed parameters of atomic model building are provided in Table 2.

## Reporting summary

Further information on research design is available in the Nature Portfolio Reporting Summary linked to this article.

## Data availability

Unless otherwise stated, all data supporting the results of this study can be found in the article, supplementary, and source data files. NMR resonance assignments for D10-NT and D2-NT are available in the BMRB databank under accession numbers 52550 and 52551, respectively. PDB files are available in the Protein Data Bank (PDB) under accession numbers: 9CWW (D10-NT polymorph-1), 9OYQ (D10-NT polymorph-2), 9OYW (D10-NT polymorph-3), 9OYR (D2-NT), 9OYS (S59L D10-NT), 9OYT (T61I D2-NT), and 9OYO (R15L D10-NT) and cryo-EM maps are available in the Electron Microscopy Data Bank (EMDB) under accession numbers: EMD-45976 (D10-NT polymorph-1), EMD-71031 (D10-NT polymorph-2), EMD-71037 (D10-NT polymorph-3), EMD-71032 (D2-NT), EMD-71033 (S59L D10-NT), EMD-71034 (T61I D2-NT), and EMD-71028 (R15L D10-NT). TEM and cryo-EM micrographs and confocal microscopy images are available at FigShare, [https://doi.org/10.6084/m9.figshare.26516104]. Raw CD, Secondary Chemical Shift, ThT fluorescence, ThT T50, NMR intensity ratio, and FSC curve data, as well as the uncropped sarcosyl extract filter trap blot are provided in the Source Data file. Source data are provided with this paper.

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

## Acknowledgements

This work was funded in part by National Institutes of Health (NIH) grants R01NS139141 (H.K. and D.E.), RF1AG066493 (D.E.), R35NS122209 (G.M.), U01NS134684 (N.A.S.), F31AG077836 (C.M.) and F31HL154651 (N.M.S.), by MDA grant 961871-02 (H.K.), and by Project ALS grant 2021-01 (H.K.). WCM core facilities are supported by NIH shared instrumentation grants S10OD028556 and S10OD016320 (NMR core), and 1S10OD026974 (Neuroanatomy Electron Microscopy core, and S10RR027699 (Electron Microscopy & Histology services of the Weill Cornell Medicine Microscopy & Image Analysis core). Some of the work was performed at the WCM Cryo-EM Core Facility, at the NYU Langone Health's Cryo-Electron Microscopy Laboratory (RRID: SCR_019202), which is partially supported by the Laura and Isaac Perlmutter Cancer Center Support Grant NIH/NCI P30CA016087, and at the Simons Electron Microscopy Center at the New York Structural Biology Center (NYSBC), with major support from the Simons Foundation (SF349247). Data collected using the 800-MHz Avance III spectrometer is supported by NIH grant S10OD016432 (NYSBC). The 5mm 800-MHz TXO cryoprobe is supported by NIH grant S10OD028577 (NYSBC). We acknowledge assistance from Clay Bracken and Emily Grasso with NMR data collection and processing, from Teri Milner and Lee Cohen-Gould with negative stain and immunogold EM data collection, from Carl Fluck with cryo-EM data collection, from Biao Qiu, Carl Fluck, Greg Alushin and Sjors Scheres with cryo-EM data processing and from Nicole Casey with the production and execution of full length D2/D10 in-vitro fibrilization experiments.

## Author contributions

D.E., H.K and G.M. conceived and designed the project; G.L. and Y.H. carried out the cryo-EM studies. G.L., N.M.S., and C.M. produced recombinant proteins and carried out the CD studies and ThT assays. C.M. carried out the NMR studies. GL carried out the TEM studies. N.M.S., G.M., H.K. and G.L. carried out the immunogold EM studies. N.M.S. performed the sarcosyl extraction. N.M.S. and H.K. carried out the mouse tissue filter trap assays. K.M., N.M.S., G.M. and H.K. carried out

the fibroblast imaging studies. N.A.S. provided the patient fibroblasts. D.E., H.K., G.M., G.L., N.M.S., Y.H., C.M., K.M., and N.A.S. wrote and revised the paper.

## Competing interests

The authors declare no competing interests.
