## [Transparent Peer Review file · Nature Communications]

Amyloid fibril structures link CHCHD10 and CHCHD2 to neurodegeneration

Corresponding Author: Dr David Eliezer

Version 1:

Reviewer comments:

Reviewer #1

(Remarks to the Author)

The manuscript by Lv et al from the Eliezer group entitled "Amyloid fibril structures link CHCHD10 and CHCHD2 to neurodegeneration" determined several cryo EM structures (i.e. D10-NT, D10-NT S59L and the D2-NT) of amyloids of a fragment of CHCHD10 and CHCHD2 related to PD, FTD or ALS. Further they correlated their findings to the sarkosyl insoluble fraction of heart tissue from a D10S55L mouse and to patient derived fibroblast cells containing the D10 R15L mutation. For a detailed understanding on the complex nature of the age-related neurodegenerative diseases it is highly important to enlarge the study of the pool of proteins involved and structures with it. This is here done with the amyloid structure of fragments of the two proteins studied. However, the presented work is not yet complete as the link between structure and co-aggregation or disease is not established yet. The following points are requested to be considered:

(i) The introduction is too short. The protein subjects and their relationship to disease should be introduced in more details.

(ii) The link between in vivo observation and the fibril structure is not coherent as other mutants have been studied in vivo than in the in vitro analysis (in addition to the full-length protein versus N-terminal fragments thereof). In details, the R15L mutant over the S59L mutation for the study with the patient derived fibroblast cell line shown in figure 1D is used. Neither in the ThT kinetics nor in cryo-EM studies the R15L mutation is used. It would be a great support of the manuscript to be more coherent. This should be resolved eventually by determining the amyloid structures of the in vivo mutant fragments R15L.

(iii) The correlation between mutant versus wild-type CHCHD10 and between CHCHD10 and CHCHD2 requests cross seeding experiments along with structure determination. Can CHCHD10 efficiently cross seed CHCHD2 without perturbing or by perturbing the polymorph structure? Can mutant CHCD10 cross seed wild-type without perturbing or by perturbing the polymorph structure. With the structures in hand having one protofilament similar to each other, while the other one not, it is indicated that the mutant cross-seeds with wild-type preserving its structure. Similar with Figure 5 it is indicated that CHCHD10 may cross seed CHCHD2 without perturbing the polymorph structure of the latter.

(iv) The question is further raised whether a-syn amyloids are able to cross seed with CHCHD10 and if yes, whether again the polymorph is perturbed or not.

It is the point of the reviewer that such structure-activity relationship studies [listed under (iii) and (iv)] will be highly exciting towards understanding the complex nature of aggregation in a neurodegenerative organism specifically within the subject of interest here (CHCHD2/10) and in a broader context.

(v) Another important question is whether the fibril structure formations are reproducible. It is also noted that the aggregation conditions differed (i.e. different concentration and different time) between the cryo EM investigations asking the question how the conditions influence the polymorph structure.

(vi) In Fig 1E the sarcosyl-insoluble fraction of the D10 S55L mouse heart tissue are screened using immuno-gold nanoparticles. In comparison to other studies (<https://doi.org/10.1186/s40478-023-01658-y>), the amount of gold nanoparticles across the fibril is not convincing, it would be great if additional images of the D10 S55L fibrils could support the hypothesis.

Minor points:

(vii) In Fig. 1C the fibrils of D10-NT of the negative stain TEM image correspond to which point (8h or after 5d of aggregation)? Further, the fibril in Fig. 1C show multiple types of diameters, it would be interesting to understand if that correlates with the cryo-EM data. Therefore, it is suggest to run initial 2D Classification over all the extracted particle for each of the data sets. I suggest to use for the extraction job a box size of 512 and pixel size of 2x 1.076 Å (for D10-NT and S59L D10-NT) and for D2-NT a box size of 512 and 3x 0.825 Å. For the 2D Classification around 500 Classes should be used and then the new filament subset selection tool (Select subsets of filaments using dendrograms, <https://github.com/dbli2000/FilamentTools>) from relion5 can be used to understand the polymorph landscape or loss of particles during the classification process.

(viii) In the method part it is specified that the ThT measurement is done in triplicates. It is thus assumed that a single time point is measured three times. Why does the error bar for 4h D10-NT is significant smaller, than for all the other measurements? Further, is this aggregation behavior in some way reproducible, that the kinetics for S59L D10-NT are always faster than wt.

(ix) In the Cryo-EM method section some details are missing. The authors in general specify the box size, but do not mention the corresponding pixel size, please add the pixel size to the following parts:

“Fibrils were manually picked and segments were extracted using a large box size, typically 768 pixels.”

“Further re-extraction to a smaller box size, typically 384 pixels, and 2D classification was conducted and the resulting 2D classes were used to generate initial models using the `relion_helix_inimodel2d` program.”

“Segments were re-extracted to smaller box size (256 pixels) and subjected to 3D auto-refinement without a mask.”

Additionally, it would be helpful to specify the crossover distance used in the initial model building with `relion_helix_inimodel2d` and show for each structure the initial model and the corresponding cross section.

The scale bars in certain figures are missing (Fig 1 C-D, Fig. S2C) or are too small and difficult to read (Fig. 3A, Fig. 4A, Fig. S3A). Further it would be great to make the scale bar always the same length, which makes it easier to compare for the reader.

Reviewer #2

(Remarks to the Author)

This study shows that recombinant proteins corresponding to the predicted N-terminal low-complexity regions of CHCHD10 and CHCHD2 undergo fibrillization. The pathogenic CHCHD10 S59L mutation, as shown in previous studies, exhibits high aggregation propensity and forms fibrils with a morphology distinct from that of the wild-type. In contrast, the fibril core of CHCHD2 differs from that of CHCHD10. Based on these observations, the authors suggest that neurodegenerative diseases associated with CHCHD10 and CHCHD2 may be classified as amyloid fibril-type neurodegenerative diseases.

While this hypothesis is entirely novel in the relevant research field, three major issues need to be addressed:

1. As acknowledged by the authors, one significant limitation of this study is the lack of *in vivo* evidence demonstrating fibrillization. The fibrillization observed *in vitro* at a high concentration of 16 μ M raises questions about whether this phenomenon can occur physiologically. Furthermore, the study does not explain why wild-type CHCHD10 underwent fibrillization *in vitro*, whereas in CHCHD10 mouse models, aggregation was detected only with the S55L mutation. Additionally, it remains unclear whether full-length proteins containing the CHCH domain can also undergo fibrillization.
2. Pathogenic mutations also exist outside the fibril core. This point should at least be mentioned in the manuscript.
3. CHCHD2 and CHCHD10 physiologically form heterodimers, yet the possibility of mixed fibrils between CHCHD2 and CHCHD10 has been overlooked.

Other comments:

The CHCHD2 T61I mutation is known to exhibit high aggregation propensity, similar to CHCHD10 S59L. Based on the fibril core structure of CHCHD2 presented in Fig. 4E, the T61I mutation is expected to have a significant impact on the fibril structure. Data on the fibrils of T61I should also be presented.

The Thioflavin staining in Fig. 1D shows poor correlation with the CHCHD10 signal, raising concerns about its reliability. Additionally, the differences in the Thioflavin background between the control and R15L images are problematic. Furthermore, the images lack scale bars.

Reviewer #3

(Remarks to the Author)

Version 2:

Reviewer comments:

Reviewer #1

(Remarks to the Author)

The revised version of the manuscript entitled "Amyloid fibril structures link CHCHD10 and CHCHD2 to neurodegeneration" by Lv et al. is much improved also because several further cryo EM structures including mutants that link the in vitro data with the in vivo data have been determined. There are however two important points that request further considerations:

(i) Since the coaggregation/coseeding/hetero-seeding could not be demonstrated (not added in the manuscript and only mentioned in the answers to reviewers section) the claims/statement/passages around it should be strongly reduced and softened in the text. For example, in the abstract, there is the statement "a subset of the CHCHD10 and CHCHD2 fibril structures appear intercompatible and could accommodate sequence variations between the two proteins, suggesting that the observed co-deposition and co-aggregation of the two proteins may occur at the level of mixed amyloid fibrils" while there is no data on it in the manuscript. There is only a structural comparison. Also in the case of other disease associated amyloids there are common structural motifs between different polymorphs but that does not mean yet hetero-seeding and/other mixed fibril formation.

(ii) The reading by the reviewer of the table 1 suggests that with two samples with wt D10-NT three polymorph structures were solved meaning that no reproducibility is demonstrated. With this lack of reproducibility the mutant influence on the structures may not be meaningful at all and thus should not be done. Reproducibility, that may include a variation of distributions of several polymorphs is o.k., but only then it can be stated that a mutant polymorph structure is due to the mutation and not just because of reproducibility issues. With other words if a discussion on the influence on a specific mutation on the polymorph structure is done one should be able to exclude that the same polymorph is not formed by the wt (or formed but with a different propensity). The reviewer understands the work load that this requests, but there appears no other way if one wants to state that the mutation has an influence on the structure.

The authors have addressed many of the previously open questions and the revised manuscript is much improved. However, as known from other pathological amyloids such as tau, Abeta, and synuclein small sample differences including pH and buffer composition can significantly influence polymorphism. These variables should be more explicitly taken into account in the manuscript.

For example, the authors state:

"We solve the structures of N-terminal domain fibrils using cryogenic electron microscopy (cryo-EM) and show that they are comprised of the most highly conserved region between the two proteins. We further show that disease-associated mutations within this region result in altered fibril structures, but that sequence differences between D10 and D2 in this region could likely be accommodated in mixed D10/D2 fibrils. Together, our findings strongly implicate amyloid fibril formation in the etiology of D10 and D2 associated neurodegenerative disease."

The authors also present convincing biochemical data using sarkosyl-insoluble fractions, including filter trap assays, dot blots (comparing WT D10 mice and D10 S55L mutants), and immunogold staining of D10 S55L fibrils analyzed by TEM. These experiments support the presence of aggregates in the mutant but not in the wild-type tissue. They use this to justify a functional correlation between the human and mouse sequences:

"Mouse D10 is highly homologous to human D10 (Fig. 3A), with serine residue 55 corresponding to human residue S59, which is mutated to leucine in disease. The presence of D10 aggregates in these extracts, but not in extracts from WT mice (D10WT), was confirmed using a filter trap assay (Fig. 3B)."

However, based on Figure 3A, there are more than 10 amino acid mismatches between the human and mouse D10 sequences. While the S55L mutation may indeed result in more aggregate formation in the mouse model compared to wild-type mouse, the reviewer is not fully convinced that this observation alone allows for a direct correlation with the human pathological sequence. As the authors themselves note:

"We also observe evidence for WT D10 aggregation in filter blots of very old (P500+) mice (Supp. Fig. 281 S10). At present, there are no published reports associating D10 and D2 aggregation with disease in the absence of D10 or D2 mutations, but our results suggest that such an association is possible and should be investigated."

It is noteworthy that the authors report three different polymorphs for human D10, while D10 S59L gives rise to a single asymmetric polymorph. They argue that this is due to the point mutation. The reviewer agrees that a "serine to leucine" substitution could impact fibril structure. However, if experiments are repeated multiple times, it is rather expected that a broader polymorph landscape will emerge. This would align with the stochastic and heterogeneous nature of fibril formation, which makes it challenging to draw conclusions such as:

"However, disease-linked mutations are also found outside the fibril core regions. As we show here for the p.R15L D10 mutation, such mutations are less likely to influence fibril structure but may still alter the fibril assembly process. Again, other known amyloids also feature disease-linked mutations outside their fibril core regions (e.g. p.R5H for tau41, p.A30P for alpha synuclein)."

In summary, while the manuscript convincingly demonstrates that the point mutation influences polymorphism, the broader context of polymorph variability, likely influenced by multiple experimental parameters should be acknowledged. As it

stands, the manuscript may give the impression that fibril polymorphism is dictated solely by the mutation, whereas it is more likely to arise from a complex interplay of factors and lack of sufficient repetition of structure determination.

Reviewer #2

(Remarks to the Author)

In light of the authors' response, I have no further concerns. However, it would be helpful for peer reviewers in the future if you could clearly indicate the individual parts that have been revised in your rebuttal letter.

Reviewer #3

(Remarks to the Author)

Version 3:

Reviewer comments:

Reviewer #1

(Remarks to the Author)

The authors have answered all questions and request of the reviewer. The only request left is stating in the manuscript whether the new cryo EM data have been measured with the same protein preparations or another ones.

Reviewer #3

(Remarks to the Author)

Weill Cornell Medicine

David Eliezer, PhD
Professor of Biochemistry

1300 York Avenue Box 63
New York, NY 10065-4896

dae2005@med.cornell.edu

Department of Biochemistry

March 24, 2025

Response to referee comments:

We appreciate the time and care taken by the referees in evaluating our original manuscript. We have improved our manuscript according to their comments and provide a detailed point-by-point response to each of their critiques below. Notably, we have determined and included 4 additional high-resolution cryo-EM structures in the revised manuscript, as per the referees' suggestions, and added or improved additional data items.

Reviewer #1 (Remarks to the Author):

The manuscript by Lv et al from the Eliezer group entitled "Amyloid fibril structures link CHCHD10 and CHCHD2 to neurodegeneration" determined several cryo EM structures (i.e. D10-NT, D10-NT S59L and the D2-NT) of amyloids of a fragment of CHCHD10 and CHCHD2 related to PD, FTD or ALS. Further they correlated their findings to the sarkosyl insoluble fraction of heart tissue from a D10S55L mouse and to patient derived fibroblast cells containing the D10 R15L mutation. For a detailed understanding on the complex nature of the age-related neurodegenerative diseases it is highly important to enlarge the study of the pool of proteins involved and structures with it. This is here done with the amyloid structure of fragments of the two proteins studied.

We thank the referee for noting the value and importance of our D10-NT and D2-NT amyloid structures for enlarging the study of the pool of proteins involved in neurodegenerative disease and of their structures.

However, the presented work is not yet complete as the link between structure and co-aggregation or disease is not established yet. The following points are requested to be considered:

(i) The introduction is too short. The protein subjects and their relationship to disease should be introduced in more details.

We have reformatted our manuscript from the original communication format to the more traditional article format and have included a more thorough introduction. A version of the manuscript highlighting all substantial changes is include for the referees' convenience.

(ii) The link between in vivo observation and the fibril structure is not coherent as other mutants have been studied in vivo than in the in vitro analysis (in addition to the full-length protein versus N-terminal fragments thereof). In details, the R15L mutant over the S59L mutation for the study with the patient derived fibroblast cell line shown in figure 1D is used. Neither in the ThT kinetics nor in cryo-EM studies the R15L mutation is used. It would be a great support of the manuscript to be more coherent. This should be resolved eventually by determining the amyloid structures of the in vivo mutant fragments R15L.

The referee correctly points out that our analysis of D10 amyloid formation in patient fibroblasts used cells from a patient with the R15L mutation, while our structural studies of a D10 disease mutant were for the S59L mutation. We have now added structural studies as well as ThT aggregation kinetics for the R15L mutant, as requested by the referee, to increase the correspondence between the in vitro and in vivo studies.

(iii) The correlation between mutant versus wild-type CHCHD10 and between CHCHD10 and CHCHD2 requests cross seeding experiments along with structure determination. Can CHCHD10 efficiently cross seed CHCHD2 without perturbing or by perturbing the polymorph structure? Can mutant CHCD10 cross seed wild-type without perturbing or by perturbing the polymorph structure. With the structures in hand having one protofilament similar to each other, while the other one not, it is indicated that the mutant cross-seeds with wild-type preserving its structure. Similar with Figure 5 it is indicated that CHCHD10 may cross seed CHCHD2 without perturbing the polymorph structure of the latter.

The referee points out that our structures suggest potential compatibility between the CHCHD10 and CHCHD2 fibril structures, suggesting that cross-seeding between the two proteins is possible, and similarly that cross-seeding between WT and S59L D10 may be possible. These are exciting possibilities that warrant further investigation. Indeed, we have attempted both cross-seeding reactions, but to date, we have not observed evidence of cross-seeding or of mixed fibril formation. We note that this may be due to quite a number of variables, including the specific structure adopted by the seeds and the conditions of the reactions, and thoroughly exploring these variables is a major undertaking that is beyond the scope of the current work. We do highlight the potential for cross-seeding and mixed fibril formation between D10 and D2 in the manuscript and note that it may explain the observed co-deposition of the proteins.

(iv) The question is further raised whether a-syn amyloids are able to cross seed with CHCHD10 and if yes, whether again the polymorph is perturbed or not. It is the point of the reviewer that such structure-activity relationship studies [listed under (iii) and (iv)] will be highly exciting towards understanding the

complex nature of aggregation in a neurodegenerative organism specifically within the subject of interest here (CHCHD2/10) and in a broader context.

The referee points out that alpha-synuclein could also potentially cross-seed with D10, or more likely D2, given reports in the literature of co-deposition of these proteins in autopsy material. This is a fascinating direction to explore in future work, and we now note this possibility in our discussion. The referee is certainly correct that such studies would be highly exciting and would expand our understanding of D10 and D2 aggregation in disease, However, such studies are not seminal to the main message of our paper, which is the novel observation that D10 and D2 form amyloid fibrils that are likely linked to neurodegeneration, and are considerably beyond the scope of the current work.

(v) Another important question is whether the fibril structure formations are reproducible. It is also noted that the aggregation conditions differed (i.e. different concentration and different time) between the cryo EM investigations asking the question how the conditions influence the polymorph structure.

The referee notes that fibrils may take on different structures depending on experimental conditions and notes that our different aggregation reactions were not performed under identical protein concentrations and were not sampled at identical times. Our goal in this manuscript was to evaluate whether D10 and D2 can form amyloid fibrils and if so, to elucidate their structures in order to assess the potential relevance of such fibrils to disease. As such, we performed aggregation reactions under various conditions and sampled them at different times to obtain samples that were optimal for our structural studies. Different protein concentrations and aggregation time points yielded variable fibril quantity and quality, degree of fibril clumping, and presence twisted fibrils (which are required for cryo-EM structure determination).

The referee also inquires regarding the reproducibility of the fibril structures and below also asks regarding fibril polymorphs. In the revised manuscript, we provide information on the extent of reproducibility that we have observed, include the structures of two additional polymorphs of D10, each exhibiting a different structure, and discuss the nature of this heterogeneity, which is commonly observed for in vitro assembled fibrils. A thorough exploration of the effects of conditions and aggregation time on D10 and D2 in vitro fibril assembly, following work that has been reported by Lovestam for tau, for example, would be an interesting future direction, but is beyond the scope of the current work.

(vi) In Fig 1E the sarcosyl-insoluble fraction of the D10 S55L mouse heart tissue are screened using immuno-gold nanoparticles. In comparison to other studies (<https://doi.org/10.1186/s40478-023-01658-y>), the amount of gold nanoparticles across the fibril is not convincing, it would be great if additional images of the D10 S55L fibrils could support the hypothesis.

The referee contrasts model mice with images of immunogold labeled tau fibrils extracted from tau model mice and notes that our immunogold labeled sarcosyl-insoluble fibrillar material from D10 S55L model mice exhibits sparse gold particle labeling compared with that observed for tau fibril extracted from tau model mice, and suggests we acquire additional images. We note that immunogold labeling can be highly variable, depending on conditions and most especially on the location of the antibody epitope. In the tau data the referee cites, the epitope for the antibody used (BR134) recognizes the C-terminal region of tau, which is not typically involved in the core of tau amyloid fibrils and would therefore be expected to be highly accessible to the antibody. Indeed, in the corresponding images,

the gold particles are situated away from the fibril core, suggesting that they are attached to flexible 'fuzzy coat' formed by the C-terminal and N-terminal tau domains. In the case of our data, the epitope for the antibody we use is not provided, but our gold particles are found in close proximity to the fibril core, suggesting that the epitope may be within, or closely associated with, the ordered fibril core. We also observe an apparent preference of the gold particles for the ends of our fibrils, consistent with greater access to a fibril core epitope in these locations. Thus, we believe our data strongly support our hypothesis that D10 forms amyloid fibrils in vivo. As per the referee's suggestion, we have also provided additional images of sarkosyl extracts from D10 S55L mice. Furthermore, we have performed immunogold labeling of in vitro assembled D10-NT fibrils using the same antibody. The images, which we now include, exhibit a similar degree of labeling to that observed for the material obtained from the model mice, further supporting the conclusion that our immunogold labeling images reflect D10 amyloid fibril formation in vivo.

Minor points:

(vii) In Fig. 1C the fibrils of D10-NT of the negative stain TEM image correspond to which point (8h or after 5d of aggregation)? Further, the fibril in Fig. 1C show multiple types of diameters, it would be interesting to understand if that correlates with the cryo-EM data. Therefore, it is suggested to run initial 2D Classification over all the extracted particle for each of the data sets. I suggest to use for the extraction job a box size of 512 and pixel size of 2x 1.076 A (for D10-NT and S59L D10-NT) and for D2-NT a box size of 512 and 3x 0.825 A. For the 2D Classification around 500 Classes should be used and then the new filament subset selection tool (Select subsets of filaments using dendrograms, <https://github.com/dbli2000/FilamentTools>) from relion5 can be used to understand the polymorph landscape or loss of particles during the classification process.

The referee notes heterogeneity in the fibrils imaged by TEM and asks regarding the conditions for this sample and the implications for polymorph heterogeneity, suggesting a specific analysis. We have provided the sample conditions for this TEM image. This sample was from one of our earliest aggregation reactions and was not used for cryo-EM experiments.

As suggested by the referee, we have now repeated 2D classification of all our data sets using a large box size for particle extraction, followed by filament subset selection. We note that even with 50-200 classes, we still typically had many empty classes, so we did not employ 500 classes, as the referee suggested. Filament clusters were first selected by flattening the dendrograms at a specified threshold for each individual data set. Additional 2D classifications were then carried out for each cluster to assess the associated polymorph. Many clusters consisted of small numbers (<2000) or particles, of straight segments, or of clearly unsolvable classes and these were not pursued. All samples featured either one or two major clusters corresponding to distinct structures/polymorphs, except for the D2-NT sample, for which 4 different clusters appear to contain the same polymorph, although we have not been able to determine this structure. This analysis confirms that some of our preparations contain more than one polymorph and we include the details of this analysis in the table below for the referee, but in the revised manuscript we merely provide a summary of these results (new Table 1), as we believe the details will not add to the paper. To explore the polymorph landscape at high resolution, we have now determined and added the structures of two additional WT D10-NT polymorphs to the manuscript, as described above, and we identify and discuss structural similarities and differences between these different structures. There remain polymorphs that we have, to date, been unable to adequately resolve. We include a description of the number of identified polymorphs and the number of

those for which we have solved structures in a new Table 1. We note that polymorphism is often observed for *in vitro* assembled structures, and emphasize, as we did in the original manuscript, that despite potential differences between *in vitro* and *ex vivo* fibril structures, certain structural elements are often conserved in both. Indeed, we identify a specific structural element, a central VAV beta-strand motif, that is conserved in all our structures, and that we propose may therefore likely feature in *ex vivo* structures as well.

Summary of FilamentTools analysis of polymorph classes.

Sample (Pixel size)	Box size in pixels	Total Extracted Particles	Classified Particles ⁺	Non- empty /Total Classes	Threshold/ Clusters	Fraction of particles with twist/number of clusters, for each polymorph	Fraction of untwisted/bad particles
D10-NT sample-1 (1.076 Å)	512 (binned to 256)	2,813,024	281,303	121/200	0.76/27	0.36/1* 0.087/1*	0.553
D10-NT sample-2 (1.16 Å)	784 (binned to 392)	415,983 [#]	415,983	5/100	0.7/2	0.982/1*	0.018
D2-NT (0.825 Å)	512 (binned to 256)	2,271,215	227,122	199/200	0.85/111	0.1711/1* 0.0829/4	0.746
S59L D10-NT (1.076 Å)	512 (not binned)	658,614 [#]	658,614	28/50	0.74/6	0.7465/1* 0.1933/1	0.0602
T61I D2-NT (1.16 Å)	512 (binned to 256)	1,132,117	566,058	77/200	0.8/18	0.7662/1*	0.2338
R15L D10-NT (1.083 Å)	512 (binned to 256)	2,184,538	218,454	46/200	0.82/6	0.9991/1*	0.0009

[#]These data sets contained fewer particles and were originally extracted with a sufficiently large box size of (784 or 512 pixels), and therefore did not require re-extraction for this analysis.

⁺For data sets contain more than 1 million particles and FilamentTools fails on our cluster when using all the particles. Therefore, randomly selected subsets of all particles were included in analyzing these samples using FilamentTools.

*Polymorphs for which we were able to determine a structure.

(viii) In the method part it is specified that the ThT measurement is done in triplicates. It is thus assumed that a single time point is measured three times. Why does the error bar for 4h D10-NT is significant smaller, than for all the other measurements? Further, is this aggregation behavior in some way reproducible, that the kinetics for S59L D10-NT are always faster than wt.

All of the ThT data presented were measured three times at each time point, as the referee assumes. The variability between the three measurements is itself variable and the smaller error bar for the 4-hour time point simply reflects the fact that these three measured values were closer to each other than for other time points. This is also true for the 1-hour time point as well as earlier time points. For clarity, we now include all the data points in the plot, in addition to the average.

The referee asks regarding the reproducibility of the difference between WT and S59L D10-NT aggregation kinetics. We have found the dramatic difference in these aggregation kinetics to be highly reproducible, S59L D10-NT always aggregates much more quickly. We provide further evidence in support of the faster aggregation of S59L D10-NT in the form of NMR data which show that even

without agitation, S59L is prone to inter-molecular interactions that lead to a broadening of the NMR signals from the region of the protein that includes the fibril core. We also observe this phenomena (broadening of NMR signals from the fibril core) for the R15L D10-NT mutant, for which we now include both a structure as well as ThT aggregation kinetics indicating that this mutant tends to aggregate more rapidly, albeit only slightly, than the WT. Interestingly, the T61I D2-NT mutant, for which we also now include a structure and ThT aggregation kinetics, does not aggregate more rapidly, and does not feature NMR signal broadening. To further illustrate the reproducibility of our ThT kinetics, we include additional data here for the referee. These experiments were not performed using our standard conditions, and are therefore not included in the manuscript, but the results confirm that S59L D10-NT aggregates much more rapidly than WT (noting that in one of the two additional experiments provided here, ThT values reached their maximal value at the earliest sampled time point, precluding a determination of T_{50} , while in the other experiment, the sample volumes were insufficient to acquire the later time points, precluding statistical analyses of these results), that R15L D10-NT aggregates slightly faster than WT, that D2-NT aggregates more slowly than WT, and that T61I D2-NT aggregation rates do not differ from those of WT D2-NT.

(ix) In the Cryo-EM method section some details are missing. The authors in general specify the box size, but do not mention the corresponding pixel size, please add the pixel size to the following parts: “Fibrils were manually picked and segments were extracted using a large box size, typically 768 pixels.”

“Further re-extraction to a smaller box size, typically 384 pixels, and 2D classification was conducted and the resulting 2D classes were used to generate initial models using the `relion_helix_inimodel2d` program.”

“Segments were re-extracted to smaller box size (256 pixels) and subjected to 3D auto-refinement without a mask.”

Additionally, it would be helpful to specify the crossover distance used in the initial model building with `relion_helix_inimodel2d` and show for each structure the initial model and the corresponding cross section.

The scale bars in certain figures are missing (Fig 1 C-D, Fig. S2C) or are too small and difficult to read (Fig. 3A, Fig. 4A, Fig. S3A). Further it would be great to make the scale bar always the same length, which makes it easier to compare for the reader.

The referee rightly requests that we include information regarding pixel size in the methods description of the cryo-EM data analysis. We apologize for this omission and now include this information for each data set both in Table 2 and in the supplementary figures associated with each structure. We also include the requested information regarding crossover distance and show for each structure the initial model and cross section in a new supplementary figure, as requested. We have also improved the size and uniformity of the scale bars in all images.

Reviewer #2 (Remarks to the Author):

This study shows that recombinant proteins corresponding to the predicted N-terminal low-complexity regions of CHCHD10 and CHCHD2 undergo fibrillization. The pathogenic CHCHD10 S59L mutation, as shown in previous studies, exhibits high aggregation propensity and forms fibrils with a morphology distinct from that of the wild-type. In contrast, the fibril core of CHCHD2 differs from that of CHCHD10. Based on these observations, the authors suggest that neurodegenerative diseases associated with CHCHD10 and CHCHD2 may be classified as amyloid fibril-type neurodegenerative diseases.

While this hypothesis is entirely novel in the relevant research field, three major issues need to be addressed:

We thank the referee for noting the novelty of our discoveries linking CHCHD10 and CHCHD2 amyloid fibril formation to neurodegenerative disease and address the issues they raise below.

1. As acknowledged by the authors, one significant limitation of this study is the lack of *in vivo* evidence demonstrating fibrillization. The fibrillization observed *in vitro* at a high concentration of 16 μM raises questions about whether this phenomenon can occur physiologically. Furthermore, the study does not explain why wild-type CHCHD10 underwent fibrillization *in vitro*, whereas in CHCHD10 mouse models, aggregation was detected only with the S55L mutation. Additionally, it remains unclear whether full-length proteins containing the CHCH domain can also undergo fibrillization.

The referee cites a lack of in vivo evidence demonstrating fibrillization. We note that we provide two lines of evidence for in vivo fibril formation, co-localization of immunolabeled D10 puncta with staining by the amyloid-selective dye Thioflavin-S in fibroblasts derived from human D10 patients carrying the R15L mutation, and immunogold labeling of fibrillar species isolated from the tissues of knock in D10^{S55L} mouse models. Combined, these data provide compelling evidence for amyloid fibril formation by D10 in vivo.

The referee questions why aggregation is observed for the WT protein in vitro, but detected only in the D10^{S55} mice. As we note in the manuscript, our observation that WT D10 can form fibrils in vitro suggests that aggregation of the WT protein could also occur in disease, as is commonly seen for other amyloids such as alpha-synuclein. However, this typically does not occur in normal individuals, and is instead most commonly associated with increased levels of WT protein expression (e.g. gene duplication/triplication or changes to promoter regions). While this has not been reported yet for CHCHD10 or CHCHD2, our results suggest that this warrants investigation, and this observation is another important implication of our work.

The referee also notes that our in vitro fibrillization reactions take place at a 'high' concentration of 16 μ M, which could raise questions about whether fibrillization of these proteins can occur physiologically. The concentrations we use in our in vitro reactions are actually lower than those typically used for studies of in vitro amyloid formation, which are more typically performed in the \sim 100 μ M concentration range. While the concentrations of D10 and D2 in vivo are not known, it is important to note that our in vitro studies are not intended to reproduce in vivo aggregation conditions, which is exceedingly difficult, and could also occur over impractically long time scales, but rather to demonstrate the ability of these proteins to form amyloid, and to generate material for structural studies. The relevance of the in vitro assembled fibrils to disease is supported by our observation of in vivo amyloid fibril formation, as well as the observations prompted by the resulting fibril structures (e.g. conservation of the fibril core region, placement of disease mutations in these structures and the effects of such mutations on fibril structure).

Finally, the referee asks whether full length D10 and D2, containing the CHCH domain, can also undergo fibrillization. We now include data demonstrating that both full length proteins are also able to assemble into amyloid fibrils.

2. Pathogenic mutations also exist outside the fibril core. This point should at least be mentioned in the manuscript.

The referee correctly points out that pathogenic mutations in both D10 and D2 are also found outside the fibrils core. This is now explicitly mentioned in the manuscript, as requested.

3. CHCHD2 and CHCHD10 physiologically form heterodimers, yet the possibility of mixed fibrils between CHCHD2 and CHCHD10 has been overlooked.

As indicated in our response to point iii) by Reviewer #1, our structures indeed suggest potential compatibility between the CHCHD10 and CHCHD2 fibrils, and that cross-seeding between the two proteins is possible. As noted above, we have attempted cross-seeding reactions, but to date, we have not observed evidence of cross-seeding or of mixed fibril formation. This may be due to quite a number of variables, including the specific structure adopted by the seeds and the conditions of the reactions, and thoroughly exploring these variables is a major undertaking that is beyond the scope of the current

work. We do highlight the potential for cross-seeding and mixed fibril formation between D10 and D2 in the manuscript and note that it may explain the observed co-deposition of the proteins.

Other comments:

The CHCHD2 T61I mutation is known to exhibit high aggregation propensity, similar to CHCHD10 S59L. Based on the fibril core structure of CHCHD2 presented in Fig. 4E, the T61I mutation is expected to have a significant impact on the fibril structure. Data on the fibrils of T61I should also be presented.

The referee notes that the T61I D2 mutation is also associated with D2 aggregation/deposition in disease, similar to S59L, and requests structural information for this mutant. We have now solved the structure of T61I D2-NT fibrils and included this in the revised manuscript. This mutation results in a new protofilament fold, different from that of WT D2 and of all the D10 structures we have observed, confirming that disease-associated mutants can lead to a remodeling of fibril structure. Interestingly, the T61I mutation does not lead to accelerated D2 aggregation in our in vitro assays, unlike what we observe for the S59L D10 mutation. This suggests that not all disease-associated mutations increase in vitro aggregation rates, an observation that has been made for other amyloids such as alpha-synuclein.

The Thioflavin staining in Fig. 1D shows poor correlation with the CHCHD10 signal, raising concerns about its reliability. Additionally, the differences in the Thioflavin background between the control and R15L images are problematic. Furthermore, the images lack scale bars.

The referee notes that the ThioS fluorescence in our original Figure 1D extended beyond the location of D10 immunostaining, as well as the presence of a ThioS background in the R15L patient fibroblast images, but not in the control images, and the lack of scale bars. We noted the 'halo' of ThioS fluorescence around our D10 puncta in the original manuscript and suggested that it could result from co-localization with other amyloids. However, in light of the referee's point about a higher background level in the patient vs control fibroblasts, and the known challenges of non-specific ThioS staining, we undertook new imaging of these cells. In the new images we provide, there are comparable levels of weak diffuse background ThioS staining in both patient and control cells, but bright ThioS puncta, which are numerous in patient-derived cells and rare in control cells, localize with D10 puncta. While the ThioS fluorescence still appears less punctate than the D10 signal, resulting in a 'halo' effect (that is, however, much less prominent than in our previous data), we believe this to be a result of the difference between antibody labeling and the biochemical dye as well as the small optical shift that is common when imaging two different fluorophores with different emission wavelengths.

Reviewer #3 (Remarks to the Author):

We thank referee 3 for their contributions and input into the review of article.

Sincerely,

David Eliezer

Weill Cornell Medicine

David Eliezer, PhD
Professor of Biochemistry

1300 York Avenue Box 63
New York, NY 10065-4896

dae2005@med.cornell.edu

Department of Biochemistry

May 20, 2025

Response to referee comments:

We appreciate the additional time and care taken by the referees in evaluating our revised manuscript. We have addressed their remaining comments in a detailed point-by-point below. Notably, we have 1) complied with the suggestion that we strongly reduce/soften our claims regarding mixed aggregation of D2 and D10 and 2) included additional data regarding the reproducibility of WT D10, WT D2 and S59L D10 structures, while discussing explicitly that in the absence of a more extensive survey of polymorphism and reproducibility, it is not possible to conclusively ascribe differences in structure to the effects of disease mutations.

Reviewer #1 (Remarks to the Author):

(i) Since the coaggregation/coseeding/hetero-seeding could not be demonstrated (not added in the manuscript and only mentioned in the answers to reviewers section) the claims/statement/passages around it should be strongly reduced and softened in the text. For example, in the abstract, there is the statement “a subset of the CHCHD10 and CHCHD2 fibril structures appear intercompatible and could accommodate sequence variations between the two proteins, suggesting that the observed co-deposition and co-aggregation of the two proteins may occur at the level of mixed amyloid fibrils” while there is no data on it in the manuscript. There is only a structural comparison. Also in the case of other disease associated amyloids there are common structural motifs between different polymorphs but that does not mean yet hetero-seeding and/other mixed fibril formation.

The reviewer makes a valid point that we have not demonstrated co-aggregation of D10 and D2, and have only provided structural comparisons of their fibrils. We have therefore removed the statement suggesting the potential for co-aggregation of the proteins from the abstract and restricted ourselves to the factual observations that the fibrils share a conserved core sequence and show some structural similarities. This section of the abstract now reads:

“The ordered cores of these fibrils are comprised of a region highly conserved between the two proteins, and a subset of the CHCHD10 and CHCHD2 fibril structures share structural similarities and appear compatible with sequence variations in this region.”

We have also, as suggested, reduced and softened our claims and statements regarding co-aggregation and mixed fibril formation in the results and discussion sections, and we now explicitly note that mixed fibrils are rare and that other amyloids like tau have not been shown to form mixed fibrils despite featuring homologous isoforms with polymorphs that share structural similarities. Changes are highlighted in the revised manuscript and the relevant discussion section now reads:

“However, our attempts to seed D2-NT fibrils with preassembled D10-NT fibrils did not result in a shorter D2 assembly lag phase, suggesting a lack of cross-templating under our conditions. Notably, mixed amyloid fibrils have only rarely been observed⁵³, despite the fact that other amyloids like tau feature homologous isoforms that populate polymorphs with structural similarities. Further work will be required to more thoroughly assess whether D10 and D2 can form fibrils with each other, or possibly with non-homologous associated proteins like alpha-synuclein⁵⁴, as has recently been reported for TDP-43 and annexin-A11⁵⁵.”

(ii) The reading by the reviewer of the table 1 suggests that with two samples with wt D10-NT three polymorph structures were solved meaning that no reproducibility is demonstrated. With this lack of reproducibility the mutant influence on the structures may not be meaningful at all and thus should not be done. Reproducibility, that may include a variation of distributions of several polymorphs is o.k., but only then it can be stated that a mutant polymorph structure is due to the mutation and not just because of reproducibility issues. With other words if a discussion on the influence on a specific mutation on the polymorph structure is done one should be able to exclude that the same polymorph is not formed by the wt (or formed but with a different propensity). The reviewer understands the work load that this requests, but there appears no other way if one wants to state that the mutation has an influence on the structure.

The reviewer makes a valid point that the polymorphism observed for WT D10 implies that the S59L structure (and the D2 T61I structure) could merely be another polymorph, and not result from the mutation. We have addressed this critique in three ways.

-First, we explicitly mention in the revised manuscript that the mutant structures may simply represent polymorphism and are not necessarily a unique result of the mutations

-Second, we have collected additional data sets for independently prepared WT D10-NT, S59L D10-NT and WT D2-NT fibril samples and performed 2D classification. The results demonstrate that the new WT D10-NT sample closely matches the original sample from which we determined the polymorph-2 structure, the new S59L D10-NT sample closely matches the original S59L sample from which we determined the S59L structure, and the new WT D2-NT sample closely matches the original D2-NT sample. Despite providing only one additional data point for each of these variants, which consists of only 2D classes, these data nevertheless suggest that the structures we observed originally are at least somewhat reproducible and preferred by their respective variants. In this regard, we also note that the

R15L D10-NT structure, which matches the WT D10-NT polymorph-2, adds further evidence for a preference of the WT protein for this structure when the WT core sequence is preserved.

-Third, we have revised our presentation and discussion of the mutant structures, removing language implying that the mutations directly result in different structures, and instead using softer language indicating that the mutations may promote alternative structures. These changes are highlighted in the revised manuscript and the relevant sections of the Results and Discussion sections now state:

“Although we observe different fibril structures for the S59L D10-NT and T61I D2-NT mutants, the polymorphism we observe for WT D10-NT makes it difficult to establish whether the observed mutant structures are uniquely associated with the mutants, whether the mutations alter the D10 or D2 polymorph landscape, or whether the mutant structures are simply additional polymorphs. To examine the reproducibility of the WT and mutant structures, we obtained and analyzed additional data sets for independent preparations of WT D10-NT, S59L D10-NT and WT D2-NT fibrils (see methods). For WT D10-NT, we obtained 2D classes closely resembling those observed for WT D10 polymorph-2 (Supp. Fig. S8). For S59L D10-NT, we obtained two 2D classes that closely resemble those observed in the sample that lead to the S59L D10-NT structure (Supp. Fig. S9), with similar populations. For WT D2-NT, we again obtained 2D classes highly similar to those observed in the original sample from which we solved the structure (Supp. Fig. S10).”

and

“However, our observation of multiple WT D10-NT structures suggests the possibility that WT D10-NT and D2-NT could also adopt the structures we observed for the S59L and T61I mutants. We have provided evidence in the form of 2D classes obtained from cryo-EM datasets of independent samples that the WT D10-NT polymorph-2, the S59L D10-NT and the WT D2-NT structures we determined are reproducible and may therefore be preferred by their respective variants. However, resolving the structural effects of mutants unambiguously must await a comprehensive exploration of the polymorph landscape of the WT and mutant proteins. Notably, even if mutations do not lead to unique fibril structures, they may favor or promote alternative structures compared to the WT proteins, both in vitro and in patients.”

The authors have addressed many of the previously open questions and the revised manuscript is much improved. However, as known from other pathological amyloids such as tau, Abeta, and synuclein small sample differences including pH and buffer composition can significantly influence polymorphism. These variables should be more explicitly taken into account in the manuscript.

We now explicitly point out that small sample difference in pH or buffer composition can lead to polymorphism. The relevant text in the results section is highlighted in the revised manuscript and now reads:

“Different polymorphs can result from even slight changes in sample conditions such as pH and buffer composition^{28,29}, and can in principle represent both different endpoints of an aggregation reaction as well as different time points along the aggregation process³⁰.”

For example, the authors state:

“We solve the structures of N-terminal domain fibrils using cryogenic electron microscopy (cryo-EM)

and show that they are comprised of the most highly conserved region between the two proteins. We further show that disease-associated mutations within this region result in altered fibril structures, but that sequence differences between D10 and D2 in this region could likely be accommodated in mixed D10/D2 fibrils. Together, our findings strongly implicate amyloid fibril formation in the etiology of D10 and D2 associated neurodegenerative disease."

As described above, we have reduced and softened our claims and statements regarding both the effects of mutations and co-aggregation and mixed fibril formation in the results and discussion sections. This particular section, which falls at the end of the introduction, now reads:

"We solve the structures of N-terminal domain fibrils using cryogenic electron microscopy (cryo-EM) and show that they are comprised of the most highly conserved region between the two proteins. We further show that disease-associated mutations within this region promote alternative fibril structures. Together, our findings strongly implicate amyloid fibril formation in the etiology of D10 and D2 associated neurodegenerative disease."

The authors also present convincing biochemical data using sarkosyl-insoluble fractions, including filter trap assays, dot blots (comparing WT D10 mice and D10 S55L mutants), and immunogold staining of D10 S55L fibrils analyzed by TEM. These experiments support the presence of aggregates in the mutant but not in the wild-type tissue. They use this to justify a functional correlation between the human and mouse sequences:

"Mouse D10 is highly homologous to human D10 (Fig. 3A), with serine residue 55 corresponding to human residue S59, which is mutated to leucine in disease. The presence of D10 aggregates in these extracts, but not in extracts from WT mice (D10WT), was confirmed using a filter trap assay (Fig. 3B)."

However, based on Figure 3A, there are more than 10 amino acid mismatches between the human and mouse D10 sequences. While the S55L mutation may indeed result in more aggregate formation in the mouse model compared to wild-type mouse, the reviewer is not fully convinced that this observation alone allows for a direct correlation with the human pathological sequence. As the authors themselves note:

"We also observe evidence for WT D10 aggregation in filter blots of very old (P500+) mice (Supp. Fig. 281 S10). At present, there are no published reports associating D10 and D2 aggregation with disease in the absence of D10 or D2 mutations, but our results suggest that such an association is possible and should be investigated."

The S55L D10 mouse has been carefully validated and published as a model for D10-associated disease, as we indicated in the sentence preceding the one cited above: "To further assess whether D10 forms amyloid fibrils in vivo, we extracted sarcosyl insoluble material from D10^{S55L} mice, which recapitulate D10 deposition as well as disease-associated phenotypes⁵. Mouse D10 is highly homologous to human D10 (Fig. 3A)..." While the mouse and human proteins are not identical, ~10 sequence variations over 140 residues is a very high level of sequence conservation, and residue S55/59 falls in an even more highly conserved stretch, with no sequence variations for 8 residues preceding and 11 residues following this site. Therefore, we contend there is no reasonable doubt about the robustness of the S55L model.

As for our observation of WT D10 aggregation in very old (>500 day old) WT mice, this observation simply confirms that the aggregation of the WT protein, which we observe in vitro, can also occur in vivo. This does not in any way rule out a role of S55L D10 aggregation in disease. Tau is a classic case where mutations lead to early onset of disease (FTDP-17), while aggregation of the wild type protein is linked to disease with a later onset (AD). Nevertheless, we now acknowledge that WT D10 aggregation would not necessarily be linked with disease, as follows:

“We also observe evidence for WT D10 aggregation in filter blots of very old (P500+) mice (Supp. Fig. S13). At present, there are no published reports associating D10 and D2 aggregation with disease in the absence of D10 or D2 mutations, and aggregation of the WT protein at advanced ages may occur without disease, as has been observed for Amyloid beta⁵⁰ or TMEM106B⁵¹. Nevertheless, our results suggest that such an association is possible and should be investigated.”

It is noteworthy that the authors report three different polymorphs for human D10, while D10 S59L gives rise to a single asymmetric polymorph. They argue that this is due to the point mutation. The reviewer agrees that a "serine to leucine" substitution could impact fibril structure. However, if experiments are repeated multiple times, it is rather expected that a broader polymorph landscape will emerge. This would align with the stochastic and heterogeneous nature of fibril formation, which makes it challenging to draw conclusions such as:

“However, disease-linked mutations are also found outside the fibril core regions. As we show here for the p.R15L D10 mutation, such mutations are less likely to influence fibril structure but may still alter the fibril assembly process. Again, other known amyloids also feature disease-linked mutations outside their fibril core regions (e.g. p.R5H for tau41, p.A30P for alpha synuclein).”

In summary, while the manuscript convincingly demonstrates that the point mutation influences polymorphism, the broader context of polymorph variability, likely influenced by multiple experimental parameters should be acknowledged. As it stands, the manuscript may give the impression that fibril polymorphism is dictated solely by the mutation, whereas it is more likely to arise from a complex interplay of factors and lack of sufficient repetition of structure determination.

We describe above the steps we have taken to address concerns related to the polymorphism we observe and issues of reproducibility. In addition to other changes highlighted in the text and excerpted above, the paragraph cited here by the reviewer has been revised as follows:

Disease-linked mutations are also found outside the fibril core regions of D10 and D2. Other known amyloids also feature disease-linked mutations outside their fibril core regions (e.g. p.R5H for tau⁴¹, p.A30P for alpha-synuclein⁴²). Such mutations are inherently less likely to influence fibril structure, but may still alter the fibril assembly process. We show here that the p.R15L D10 mutation, situated far outside the D10-NT fibril core region, adopts the polymorph-2 structure of WT D10-NT, providing further evidence, albeit from a single sample and structure, that this fold is favored when the WT D10-NT sequence is preserved in the fibril core region.”

Reviewer #2 (Remarks to the Author):

In light of the authors' response, I have no further concerns. However, it would be helpful for peer

reviewers in the future if you could clearly indicate the individual parts that have been revised in your rebuttal letter.

We thank the reviewer for their time and apologize for not indicating more explicitly in our previous rebuttal letter.

Reviewer #3 (Remarks to the Author):

We thank referee 3 for their contributions and input into the review of our revised article.

Sincerely,

David Eliezer

Weill Cornell Medicine

David Eliezer, PhD
Professor of Biochemistry

1300 York Avenue Box 63
New York, NY 10065-4896

dae2005@med.cornell.edu

Department of Biochemistry

June 16, 2025

Response to referee comments:

We appreciate the additional time and care taken by the referees in evaluating our further revised manuscript. We have addressed their remaining comments in a detailed point-by-point below.

Reviewer #1 (Remarks to the Author):

The authors have answered all questions and request of the reviewer. The only request left is stating in the manuscript whether the new cryo EM data have been measured with the same protein preparations or another ones.

We thank referee 1 again for their review of our revised article. We now explicitly state in the Methods describing the preparation of the samples used to demonstrate reproducibility of our structures. The relevant text now states:

Additional independent samples (independent protein preparations and aggregation reactions) of WT D10-NT, S59L D10-NT, and D2-NT were prepared as described in Supplementary Table 1 in order to evaluate reproducibility.

Sincerely,

David Eliezer